# On the Convergence and Robustness of Training GANs with Regularized Optimal Transport

**Maziar Sanjabi**
University of Southern California
sanjabi@usc.edu

**Jimmy Ba**
University of Toronto
jimmy@cs.toronto.edu

**Meisam Razaviyayn**
University of Southern California
razaviya@usc.edu

**Jason D. Lee**
University of Southern California
jasonlee@marshall.usc.edu

## Abstract

Generative Adversarial Networks (GANs) are one of the most practical methods for learning data distributions. A popular GAN formulation is based on the use of Wasserstein distance as a metric between probability distributions. Unfortunately, minimizing the Wasserstein distance between the data distribution and the generative model distribution is a computationally challenging problem as its objective is non-convex, non-smooth, and even hard to compute. In this work, we show that obtaining gradient information of the smoothed Wasserstein GAN formulation, which is based on regularized Optimal Transport (OT), is computationally effortless and hence one can apply first order optimization methods to minimize this objective. Consequently, we establish *theoretical convergence guarantee to stationarity* for a proposed class of GAN optimization algorithms. Unlike the original non-smooth formulation, our algorithm only requires solving the discriminator to approximate optimality. We apply our method to learning MNIST digits as well as CIFAR-10 images. Our experiments show that our method is computationally efficient and generates images comparable to the state of the art algorithms given the same architecture and computational power.

## 1 Introduction

Generative Adversarial Networks (GANs) have gained popularity for unsupervised learning due to their unique ability to learn the generation of realistic samples. In the absence of labels, GANs aims at finding the mapping from a known distribution, e.g. Gaussian, to an unknown data distribution, which is only represented by empirical samples. In order to measure the mapping quality, various metrics between the probability distributions have been proposed. While the original work on GANs proposed Jensen-Shannon distance [21], other works have proposed other metrics such as $f$-divergences [37, 16]. Recently, a seminal work by [4] re-surged Wasserstein distance [44] as a metric for measuring the distance between the distributions. One major advantage of Wasserstein distance, compared to Jensen-Shannon, is its continuity. In addition, Wasserstein distance is differentiable with respect to the generator parameters *almost everywhere* [4]. As a result, it is more appealing from optimization perspective. From the generator perspective, this objective function is *not* smooth with respect to the generator parameters. As we will see in this paper, this non-smoothness results in difficulties for optimization algorithms. Hence, we propose to use a smooth surrogate for the Wasserstein distance.

The introduction of Wasserstein distance as a metric for GANs re-surged the interest in the field of optimal transport [44]. [4] provided a game-representation for their proposed Wasserstein GAN

formulation based on the dual form of the resulting optimal transport problem. In this game-representation, the discriminator is comprised of a 1-Lipschitz function and aims at differentiating between real and fake samples. From optimization perspective, enforcing the Lipschitz constraint is challenging. Therefore, many heuristics, such as weight clipping [4] and gradient norm penalty [22], have been developed to impose such constraint on the discriminator. As we will see, our smoothed surrogate objective results in a natural penalizing term to softly impose various constraints such as Lipschitzness.

Studying the convergence of algorithms for optimizing GANs is an active area of research. The algorithms and analyses developed for optimizing GANs can be divided into two general categories based on the amount of effort spent on solving the discriminator problem. In the first category, which puts the same emphasis on the discriminator and the generator problem, a *simultaneous or successive* generator-discriminator (stochastic) gradient descent-ascent update is used for solving both problems. These approaches are inspired by the mirror/proximal gradient descent method which was developed for solving convex-concave games [34]. Although the GAN problem does not conform to convex-concave paradigm in general, researchers have found this procedure successful in *some special* practical GANs; and unsuccessful in some others [30]. The theoretical convergence guarantees for these methods are local and based on limiting assumptions which are typically not satisfied/verifiable in almost all practical GANs. More precisely, they either assume some (local) stability of the iterates or local/global convex-concave structure [33, 31, 14]. In all of these works, similar to our setting, some form of regularization is necessary for obtaining convergence. Compared to these methods, we do not limit the number of discriminator steps to one. But our convergence is global and only depends on the quality of the discriminator.

As opposed to the first line of work, some developed algorithm put more emphasis on the discriminator problem [32, 27, 23]. For example, [27] proved global convergence to optimality for the first order update rule on a specific problem of learning a one dimensional mixture of two Gaussians when the discriminator problem is solved to global optimality at each step. Another line of analysis, which also prioritize the discriminator problem, is based on the strategy of learning the discriminator much faster than the generator [23]. These analyses, which are inspired by the variants of the two time scale dynamic proposed by [8], do not directly require the convex-concavity of the objective function. However, they require some kind of local (global) stability which is difficult to achieve unless there is local (global) convex-concave structure. Compared to these results, our convergence analysis is agnostic to the method for solving the discriminator problem provided that the discriminator problem is solved to optimality within some accuracy. Based on our analysis, the amount of this accuracy then dictates the closeness to stationarity. Therefore, our result suggests that in order to obtain a better quality solution, it is not enough to just increase the number of steps for the generator, but one also needs to maintain high enough accuracy in the discriminator. It also suggests that the simple descent-ascent update rule might not converge– as it has also been observed before in the literature; see, e.g., [29] . Therefore, one should use algorithms similar to the two time scale approaches that give discriminator increasingly more advantage. Note that, unlike [27], we do not assume perfect discriminator which is not feasible in practice. It is also worth noting that the dual formulation of our regularized Wasserstein GAN is an unconstrained smooth convex problem in the functional domain. Therefore, it is theoretically feasible to solve it non-parametrically to any accuracy with polynomial number of steps in the functional space. To the best of our knowledge, our convergence analysis is the first result in the literature with mild assumptions that proves the global convergence to a stationary solution with polynomial number of generator steps and with approximate solutions to the discriminator at each step. Notice that this is possible thanks to the regularizer added to the discriminator problem in our formulation.

## 1.1   Related works and contributions

We study the problem of solving Wasserstein GANs from optimization perspective. In short, we solidify the intuitions that the use of regularized Wasserstein distance is beneficial in learning GANs [13, 7, 42, 18, 41] through a rigorous and novel algorithmic convergence analysis. There are three steps for obtaining such convergence guarantee.

- We prove that the *regularized* Wasserstein distance, when used in GAN problems, is smooth with respect to the generator parameters.
- We also prove that by *approximately* solving the *regularized* Wasserstein distance (discriminator steps), we can control the error in the computation of the (stochastic) gradients for the generator.

Such an error control could not be achieved for the original Wasserstein distance or other GAN formulations in general; see Proposition 3 in [9].

- Having *approximate* first order information and smoothness, we prove the convergence of vanilla stochastic gradient descent (SGD) method to a stationary solution. Our results suggests that converging to stationarity of the final solution not only depends on the number of steps in the generator, but also depends on the quality of solving the discriminator problem.

Note that our convergence result relies on the smoothness of *regularized* Wasserstein distance with respect to the *generator parameters*. The use of regularization as a means for smoothing has a long history in optimization literature [35]. In the optimal transport literature, the regularization has been used as a means to derive faster methods for computing the optimal transport. The most prominent example is *Sinkhorn distance* [13], which is based on regularizing the optimal transport problem with a negative entropy term. There are many efficient algorithms proposed for finding the Sinkhorn distance [25, 43, 3]. Recently Blondel et al. [7] noted that using strongly convex regularizers on the optimal transport problem would result in an unconstrained dual formulation which is computationally easier to solve. This unconstrained form is essential in using parametric methods, such as neural networks, for solving regularized optimal transport problems, as also observed in [42].

In [42] a regularized optimal transport with very small regularization weight is considered as an *objective* for learning GANs. Choosing a small regularization parameter is important as a strong regularization introduces bias. Although, our convergence guarantee applies to this objective, our smoothness analysis predicts that small weight regularization leads to an unstable algorithm. This fact has also been observed in our experiments.Thus, we use *Sinkhorn loss* [19, 41, 5] for which our convergence guarantee holds. We show that this loss does not introduce bias into finding the correct generative model regardless of the amount of regularization. Finally, using the insights from our theoretical analysis, we put together all the pieces of different methods for solving GANs with regularized optimal transport [42, 18, 41] to get an algorithm that is competitive both in terms of computational efficiency and quality with the state of the art methods.

## 2 Background

Given a cost function $c : \mathbb{R}^d \times \mathbb{R}^d \to \mathbb{R}$, the optimal transport cost between two distributions $\mathbf{p}$ and $\mathbf{q}$ can be defined as

$$d_c(\mathbf{p}, \mathbf{q}) = \min_{\pi \in \Pi(\mathbf{p}, \mathbf{q})} \int_{\mathcal{Y}} \int_{\mathcal{X}} \pi(x, y) c(x, y) dx dy, \tag{1}$$

where $\Pi(\mathbf{p}, \mathbf{q})$ is the set of all joint distributions having marginal distributions $\mathbf{p}$ and $\mathbf{q}$, i.e. $\int_{\mathcal{X}} \pi(x, y) dx = \mathbf{p}(y)$ and $\int_{\mathcal{Y}} \pi(x, y) dy = \mathbf{q}(x)$. Note that $\mathcal{X}$ and $\mathcal{Y}$ define the spaces of all possible $x$'s and $y$'s respectively. In practice, the distributions are replaced by their empirical samples, thus $\mathcal{X}$ and $\mathcal{Y}$ have finite supports. In such cases we still use integrals to represent finite sums. Throughout the paper, $\mathcal{X}$ and $\mathcal{Y}$ are assumed to have finite support, unless otherwise is noted.

The optimal transport cost (1) could be used as an objective for learning generative models. To be more specific, we assume that we have a base distribution $\mathbf{p}$ and among a set of parameterized family of functions $\{G_\theta, \theta \in \Theta\}$, we aim at learning a mapping $G_{\theta*}$ such that the cost $d_c(G_{\theta*}(\mathbf{q}), \mathbf{p})$ [1] is minimized. In other words the problem of generative model learning is

$$\min_{\theta \in \Theta} h_0(\theta) = d_c(G_\theta(\mathbf{q}), \mathbf{p}) = \min_{\pi \in \Pi(\mathbf{p}, \mathbf{q})} \int_{\mathcal{X}} \int_{\mathcal{Y}} \pi(x, y) \, c(G_\theta(x), y) \, dy \, dx. \tag{2}$$

### 2.1 Generative adversarial networks with $W_c$ objective

In [4] authors propose to use the dual formulation of the generative problem (2) as it is easier to parametrize the dual functions instead of the transport plan $\pi$. They call this formulation Wasserstein

GAN (WGAN). Based on Kantorovich theorem [44] the dual form of (2) could be written as

$$\min_{\theta} \ \max_{\alpha,\beta} \ \mathbb{E}_{x\sim\mathbf{p}} \, \phi_{\alpha}(G_{\theta}(x)) \ - \ \mathbb{E}_{y\sim\mathbf{q}}\psi_{\beta}(y), \tag{3}$$

$$\text{s.t.} \ \ \phi_{\alpha}(G_{\theta}(x)) - \psi_{\beta}(y) \leq c(G_{\theta}(x), y), \forall (x,y)$$

where for practical considerations we have assumed that the dual functions/discriminators, $\phi$ and $\psi$, belong to the set of parametric functions with parameters $\alpha$ and $\beta$ respectively. Note that the inner problem has a constraint over the functions $\phi$ and $\psi$. In the case where $c$ is a distance, then $\phi = \psi$ and the constraint is enforcing the 1-Lipschitz constraint on the functions $\psi = \phi$ with respect to $c$. This 1-Lipschitzness is not easy to enforce. In practice, it is usually imposed heuristically by adding some regularizer [22].

## 2.2 Regularized optimal transport

For any strongly convex function $I(\pi)$, we can define the regularized optimal transport as

$$d_{c,\lambda}(\mathbf{p}, \mathbf{q}) = \min_{\pi\in\Pi(\mathbf{p},\mathbf{q})} \ H(\pi, \theta) = \int \int \pi(x,y) \, c(x,y) \, dxdy + \lambda I(\pi). \tag{4}$$

We also define $\bar{d}_{c,\lambda}(\mathbf{p}, \mathbf{q}) = d_{c,\lambda}(\mathbf{p}, \mathbf{q}) - \lambda I(\pi^*)$, where $\pi^*$ is the optimal solution of (4). Note that, $\nabla_{\theta}d_{c,\lambda}(G_{\theta}(\mathbf{q}), \mathbf{p}) = \nabla_{\theta}\bar{d}_{c,\lambda}(G_{\theta}(\mathbf{q}), \mathbf{p})$. Among all strongly convex regularizers, the following two are the most popular ones [7]:

$$\text{KL: } I(\pi) = \int \int \pi(x,y) \log\left(\frac{\pi(x,y)}{\mathbf{q}(x)\mathbf{p}(y)}\right) dxdy \ \ \& \ \ \text{norm-2: } I(\pi) = \frac{1}{2} \int \int \frac{\pi(x,y)^2}{\mathbf{q}(x)\mathbf{p}(y)} \, dx \, dy.$$

When $c$ is a proper distance, one can show desirable properties for $d_{c,\lambda}$ and $\bar{d}_{c,\lambda}$; see [13] and Appendix A. It is also possible to prove the uniform convergence of function $d_{c,\lambda}(\mathbf{p}, \mathbf{q})$ to $d_c(\mathbf{p}, \mathbf{q})$ as $\lambda \to 0$ when $\mathbf{p}$ and $\mathbf{q}$ have finite support; see [7] and Appendix B. In the case of continuous distributions [42] proves a point-wise convergence between the two distances as $\lambda \to 0$.

## 2.3 Dual formulation for regularized optimal transport

The dual formulation for regularized optimal transport has also been covered in other recent works [42, 7]. Thus, we just present a summary of the results in Lemma 2.1 and highlight the important part in the remarks that follow; see Appendix D for a more comprehensive discussion.

**Lemma 2.1.** *Let $\phi(x)$ and $\psi(y)$ be the dual variables for the constraints in the regularized optimal transport problem (4). Let us also define the violation function $V(x,y) = \phi(x) - \psi(y) - c(x,y)$. Then, the dual of the regularized optimal transport is:*

$$d_{c,\lambda}(\mathbf{p}, \mathbf{q}) = \max_{\psi,\phi} \ \mathbb{E}_{x\sim\mathbf{q}}[\phi(x)] - \mathbb{E}_{y\sim\mathbf{p}}[\psi(y)] - \mathbb{E}_{x,y\sim\mathbf{q}\times\mathbf{p}} \left[f_{\lambda}(V(x,y))\right], \tag{5}$$

*where $f_{\lambda}(v) = \frac{\lambda}{e}e^{\frac{v}{\lambda}}$ for KL regularization and $f_{\lambda}(v) = \frac{(v_+)^2}{2\lambda}$ for norm-2 regularization. Furthermore, given the optimal dual variables $\phi$ and $\psi$, the optimal primal transport plan could be computed as $\pi(x,y) = \mathbf{q}(x)\mathbf{p}(y)M(V(x,y))$, where $M(v) = \frac{1}{e}e^{\frac{v}{\lambda}}$ for KL regularization and $M(v) = \frac{v_+}{\lambda}$ for norm-2 regularization.*

**Remark 2.1.1.** *The dual of the regularized optimal transport is a large scale unconstrained concave maximization which can be solved as one. But it is also amenable to the use of parametric method, i.e., neural networks, for representing the dual functions.*

Note that $V(x,y)$ in Lemma 2.1 represents the amount of violation from the hard constraint in the original dual formulation (3). Therefore, by adding the regularizer in the primal, we are relaxing the hard constraint in the dual representation to a soft one in the objective function. By looking at the problem from this perspective, one can find similarities between our approach and the one in [22] where the authors drop the 1-Lipschitz constraint on the discriminator and try to softly enforce it by regularizing the objective using the Jacobian of the discriminator function.

**Remark 2.1.2.** *Lemma 2.1 also provides a mapping that translates the dual solutions $\phi$ and $\psi$ to a corresponding pseudo-transport plan*

$$\pi(x,y) = \mathbf{q}(x)\mathbf{p}(y)M(V(x,y)). \tag{6}$$

*Note that although $\pi$ may not be a feasible transport plan, (6) can be used to compute an approximate gradient of the generator problem, as discussed in Section 4.*

In what follows, we assume that at each iteration of the procedure for finding the optimal generator, we have access to an oracle which solves the resulting dual of regularized optimal transport to some predefined accuracy. We say a solution $(\phi, \psi)$ is an $\epsilon$-accurate solution for (5), if the value that it achieves is within $\epsilon$ of the optimal value for (5). Such an oracle could be realized through convex optimization methods and non-parametric representations; see Appendix D.2. However, due to practical computational barriers, we opt for parametric realizations of the oracle, i.e neural networks.

## 3 Smoothness of the generative objective

Given two fixed distributions $\mathbf{q}$ and $\mathbf{p}$, let us define $h_\lambda(\theta) = d_{c,\lambda}(G_\theta(\mathbf{q}), \mathbf{p})$. In this section we prove that $h_\lambda(\theta)$ is smooth with respect to $\theta$ in contrast to the original metric $h_0(\theta)$. This is particularly important when we only solve the inner optimal transport problem within some accuracy. Due to space limitations, we only state our result on the smoothness of $h_\lambda(\theta)$ when the regularizer is KL divergence; similar result for norm-2 regularizer can be found in Theorem E.1 in the Appendix. The only difference when changing the regularizer comes from the fact that these two regularizers are strongly convex with respect to different norms.

**Theorem 3.1.** *Assume $\mathcal{X}$ and $\mathcal{Y}$ are compact, $\mathbf{p}$ and $\mathbf{q}$ have bounded entropy, $c$ is bounded from below, and there exist non-negative constants $L_1$ and $L_0$, such that:*

- *For any feasible $\theta_1, \theta_2$, $\sup_{x,y} \|\nabla_\theta c(G_{\theta_1}(x), y) - \nabla_\theta c(G_{\theta_2}(x), y)\| \le L_1 \|\theta_1 - \theta_2\|$,*

- *For any feasible $\theta_1$, $\sup_{x,y} \|\nabla_\theta c(G_{\theta_1}(x), y)\| \le L_0$,*

*then the function $h_\lambda(\theta)$ is L-Lipschitz smooth[2] with $L = L_1 + \frac{L_0^2}{\lambda}$. Moreover, for any two parameters $\theta_1$ and $\theta_2$, $\|\pi^*(\theta_1) - \pi^*(\theta_2)\|_1 \le \frac{L_0}{\lambda} \|\theta_1 - \theta_2\|$, where $\pi^*(\theta) = \arg\min_{\pi \in \Pi(\mathbf{p}, \mathbf{q})} H(\pi, \theta)$.*

**Remark 3.1.1.** *Theorem 3.1 holds for both continuous and discrete distributions. In addition, when $\mathcal{X}$ and $\mathcal{Y}$ have finite support, the entropies of $\mathbf{p}$ and $\mathbf{q}$ are automatically bounded.*

The proof of this theorem is inspired by [35] and is relegated to Appendix C, where we first prove the result for distributions with finite support, and then we extend the proof to continuous distributions. Note that unlike the non-regularized original formulation, small changes in $\theta$ results in a small change in the corresponding optimal transport plan in the regularized formulation. Consequently, after updating $\theta$, solving the inner problem would be easier as the optimal discriminator has not moved very far from the last iterate. It is also worth noticing that the assumptions of Theorem 3.1 is satisfied when the functions $c$ and $G$ are smooth and the domain of $x, y$ is compact.

## 4 Solving the generator problem to stationarity using first order methods

First order methods, including SGD and its variants such as Adam [24] or SVRG [2], are the work-horse for large scale optimization. These methods are built on top of an oracle that can generate a close approximation of the (stochastic) gradients.

Unfortunately, the original non-regularized GAN objective $h_0(\theta)$ is non-smooth. Moreover, it is impossible to obtain guaranteed good quality approximations of the its sub-gradients even if we solve the discriminator problem with high accuracy; see Proposition 3 in [9]. In contrast, we proved that the $h_\lambda(\theta)$ is smooth. Next we will prove that one can obtain decent quality estimates of its gradient by solving the corresponding regularized dual problem approximately.

**Theorem 4.1.** *Under the same assumptions as in Theorem 3.1, let $(\phi, \psi)$ be an $\epsilon$-accurate solution to the dual formulation of regularized optimal transport for a given $\theta$. Let $\pi$ be the transport plan corresponding to $(\phi, \psi)$, derived using (6). Let us also define $m(x, y) = \frac{\pi(x,y)}{\mathbf{q}(x)\mathbf{p}(y)}$ and $G = \mathbb{E}_{x,y \sim \mathbf{q} \times \mathbf{p}}\big[m(x, y)\nabla_\theta c(G_\theta(x), y)\big]$. Then,*

$$\|G - \nabla h_\lambda(\theta)\| \le \delta = O\left(\sqrt{\frac{\epsilon}{\lambda}}\right) \tag{7}$$

We only prove this result for the case where $\mathcal{X}$ and $\mathcal{Y}$ have finite support. This is the scenario that happens in practice where the distributions are replaced by their finite samples. While we believe a more general version of this result could be proved for continuous distributions by delicately repeating the same steps, such a proof goes beyond the scope of this paper. See Appendix G for the proof. Also note that it is possible to verify the quality of the discriminator/dual solutions. Due to the space limitation, we relegate the discussion on verification to Appendix D.3.

The above theorem guarantees that using the dual solver, we can generate approximate (stochastic) gradients for $h_\lambda(\theta)$. In other words, the discriminator steps in solving GANs could be viewed as a way of obtaining approximate gradient information for $h_\lambda(\theta)$. Using the above approximate (stochastic) gradients, one can provide algorithms with guaranteed convergence to approximate stationary solutions for GANs. We describe one such algorithm based on the vanilla mini-batch SGD and state its convergence guarantee.

---

**Algorithm 1** Oracle based Non-Convex SGD for GANs

---

INPUT: $\mathbf{q}$, $\mathbf{p}$, $\lambda$, $S$, $\theta_0$, $\{\alpha_t > 0\}_{t=0}^{T-1}$
**for** $t = 0, \cdots, T-1$ **do**
    Call the oracle to find $\epsilon$-approximate maximizer $(\phi_t, \psi_t)$ for the dual formulation
    Sample I.I.D. points $x_t^1, \cdots, x_t^S \sim \mathbf{q}$, $y_t^1, \cdots, y_t^S \sim \mathbf{p}$ and compute

$$g_t = \frac{1}{S^2} \sum_{i,j} \frac{\pi_t(G_\theta(x_t^i), y_t^j)}{\mathbf{q}(x_t^i)\mathbf{p}(y_t^j)} \nabla_\theta c(G_\theta(x_t^i), y_t^j)$$

    where $\pi_t$ is computed using $(\phi_t, \psi_t)$ based on (6).
    Update $\theta_{t+1} \leftarrow \theta_t - \alpha_t g_t$
**end for**

---

**Remark 4.1.1.** *In Algorithm 1, if we define $G_t = \mathbb{E}[g_t | \pi_t, \theta_t]$, then Theorem 4.1 simply states that* $\|G_t - \nabla h_\lambda(\theta_t)\| \leq \delta = O\left(\sqrt{\frac{\epsilon}{\lambda}}\right)$.

The following theorem establishes the convergence of Algorithm 1 to an approximate stationary solution of $h_\lambda$.

**Theorem 4.2.** *Let $L$ be the Lipschitz constant of the gradient of $h_\lambda$. Set $\Delta = h_\lambda(\theta_0) - \inf_\theta h_\lambda(\theta)$ and let $G_t = \mathbb{E}[g_t | \pi_t, \theta_t]$. Furthermore, assume $\|G_t - \nabla h_\lambda(\theta_t)\| \leq \delta$ and $\mathbb{E}[\|g_t - G_t\|^2 | \pi_t, \theta_t] \leq \sigma^2$, $\forall t$.*

- *If $T < \frac{2\Delta L}{\sigma^2}$, setting $\alpha_t = \frac{1}{L}$, we have $\frac{1}{T}\sum_{t=1}^{T} \mathbb{E}[\|\nabla h_\lambda(\theta_t)\|^2] \leq \frac{2L\Delta}{T} + \delta^2 + \sigma^2$.*

- *If $T \geq \frac{2\Delta L}{\sigma^2}$, setting $\alpha_t = \sqrt{\frac{2\Delta}{L\sigma^2 T}}$, we have $\frac{1}{T}\sum_{t=1}^{T} \mathbb{E}\left[\|\nabla h_\lambda(\theta_t)\|_F^2\right] \leq \sigma\sqrt{8\frac{L\Delta}{T}} + \delta^2$.*

The proof of this theorem is inspired by [20] and is presented in Appendix H.

**Remark 4.2.1.** *The second regime in Theorem 4.2 results in the following asymptotic convergence rate of expected norm of the gradient as $T \to \infty$:*

$$\min_{t=1,\cdots,T} \mathbb{E}[\|\nabla_\theta h_\lambda(\theta_t)\|^2] \leq O\left(\sqrt{\frac{L}{T}}\right) + O\left(\frac{\epsilon}{\lambda}\right). \tag{8}$$

It is worth noting that our convergence analysis also guarantees the convergence of the algorithm in [42] for generative learning which is similar to Algorithm 1.

**Remark 4.2.2.** *When the error in gradient approximation at each step $t$ is $\delta_t$, Theorem 4.2 is still valid with $\delta^2 = \frac{1}{T}\sum_{t=1}^{T} \delta_t^2$. Thus, the algorithm only needs to keep the average error in solving the inner problem small enough.*

## 4.1 Sinkhorn loss: a more robust generative objective

In practice, when using regularized Wasserstein distance $h_\lambda(\theta)$ as an objective for generative models, one needs to use very small value of $\lambda$ as noted by [42]. This is because a large $\lambda$ would introduce bias into measuring the Wasserstein distance. In fact, choosing large $\lambda$ may lead to undesired solutions; see Corollary F.0.2 for an example. A naïve approach to deal with this bias is to reduce $\lambda$. However, a reduced $\lambda$ have three dire effects: (i) based on Theorem 3.1, any change in the generator parameters would result in large changes in the optimal discriminator parameters. (ii) According to (8), smaller

$\lambda$ requires smaller error $\epsilon$ for solving discriminator to obtain the same convergence guarantee. Thus, solving the discriminator problem requires more effort. (iii) Lipschitz smoothness constant of $h_\lambda(\theta)$ increases with the decrease in $\lambda$. Thus, we have to choose smaller step-size, based on Theorem 4.2, which means slower convergence. In our experiments we observed that this situation worsens with the complexity and scale of the problem. A proposed solution in literature is to use Sinkhorn loss [19, 41] to reduce the bias in measuring the distance between the two distributions without reducing $\lambda$ and get an objective which is meaningful even for large values of $\lambda$. The Sinkhorn loss between two distributions $\mathbf{p}$ and $\mathbf{q}$ is defined as

$$L_\lambda(\mathbf{p}, \mathbf{q}) = 2\, \bar{d}_{c,\lambda}(\mathbf{p}, \mathbf{q}) - \bar{d}_{c,\lambda}(\mathbf{p}, \mathbf{p}) - \bar{d}_{c,\lambda}(\mathbf{q}, \mathbf{q}) \qquad (9)$$

In [19] Genevay et al. proved that, when $c$ is a distance, as $\lambda \to \infty$, $L_\lambda$ converges to Maximum Mean Discrepancy Distance (MMD)[15]; and when $\lambda \to 0$, $L_\lambda$ converges to $2d_c(\mathbf{p}, \mathbf{q})$. Next, we present a result that shows the robustness of $L_\lambda$ with respect to the choice of $\lambda \in (0, \infty)$ in identifying the true generator parameters; see the proof in Appendix J.

**Lemma 4.3.** *Assume $c$ is symmetric, i.e., $c(x, y) = c(y, x)$. If there exists $\theta_0$ for which $\mathbf{q} = G_{\theta_0}(\mathbf{p})$, then $\theta_0$ is a stationary solution of $L_\lambda(G_\theta(\mathbf{p}), \mathbf{q})$. Moreover, $L_\lambda(G_{\theta_0}(\mathbf{p}), \mathbf{q}) = 0$ for any $\lambda > 0$.*

Notice that the above does not hold for $d_{c,\lambda}(G_\theta(\mathbf{p}), \mathbf{q})$ unless $\lambda \to 0$. Based on the above Lemma we opt to use the following Sinkhorn loss as our generative objective

$$\min_\theta \hat{h}_\lambda(\theta) = L_\lambda(G_\theta(\mathbf{q}), \mathbf{p}) = 2\, \bar{d}_{c,\lambda}(G_\theta(\mathbf{q}), \mathbf{p}) - \bar{d}_{c,\lambda}(G_\theta(\mathbf{q}), G_\theta(\mathbf{q})) - \bar{d}_{c,\lambda}(\mathbf{p}, \mathbf{p}) \qquad (10)$$

Note that only the first two terms in $\hat{h}_\lambda(\theta)$ depend on $\theta$. To compute approximate gradients for these two terms, we need to call the discriminator oracle twice; and approximately solve $d_{c,\lambda}(G_\theta(\mathbf{q}), \mathbf{p})$ and $d_{c,\lambda}(G_\theta(\mathbf{q}), G_\theta(\mathbf{q}))$. With the returned discriminator solutions, we have two approximate gradients, one for each term. If each one of the gradients has error $\delta$, obtained by applying (7), the error in approximating the overall gradient is bounded by $\hat{\delta} = 3\, \delta$. Now if we further assume that sub-sampling would generate a stochastic gradient of variance $\sigma^2$ for each term, the overall variance of the noise in gradient would be bounded by $\hat{\sigma}^2 = 5\, \sigma^2$. We summarize the SGD based method for solving (10) in Algorithm 2 in the Appendix. With these assumptions, we can easily extend the convergence result of Theorem 4.2 to Algorithm 2.

**Corollary 4.3.1.** *By replacing $\sigma$ and $\delta$ in the convergence guarantees of Theorem 4.2 with $\hat{\sigma}$ and $\hat{\delta}$ respectively, we obtain a convergence guarantee for the SGD based method, described above, for solving the generative Sinkhorn loss optimization problem* (10).

## 5 Experiments

In this section we test a family of methods which we generally call Smoothed WGAN (SWGAN). They all use the two variants of regularized OT formulation, i.e. $h_\lambda(\theta)$ and $\hat{h}_\lambda(\theta)$, as their objective. We differentiate between the two objectives by explicitly mentioning Sinkhorn loss when it is used. We also investigate the choice of different cost functions, i.e. $L_1$ and Cosine distances. Unlike [41, 18] that solve regularized OT with a parametric approach, we use neural networks to solve the OT (discriminator) similar to [42]. In contrast with [41, 18] that use large batch-sizes to get unbiased gradient estimates, our gradient estimates are always unbiased due to the use of neural networks as discriminator. As a benchmark, we compare the SWGAN methods with the gradient penalty WGAN (WGAN-GP) [22] and other methods that uses the regularized OT objective [41, 19]. All algorithms were implemented in TensorFlow [1].

### 5.1 Learning handwritten digits

In this section we apply SWGAN methods to learn handwritten digits on the MNIST data set. Our main goal is to see the effect of different choices of objective and cost function on the performance of SWGAN methods. For details of hyper parameters and networks structures see Appendix L.3.

The first and second row of Fig. 1 corresponds to SWGAN methods with $L_1$ and Cosine cost respectively. As the SWGAN formulations allows flexibility in the choice of the cost function, we apply these costs on different representation of the images. In Fig. 1 (a), (b), (e) and (f), the cost function is applied on the pixel domain (no latent representation), while in Fig. 1 (c), (d), (g) and (h) the cost is applied on a lower dimensional latent representation of the image [41, 18], parameterized

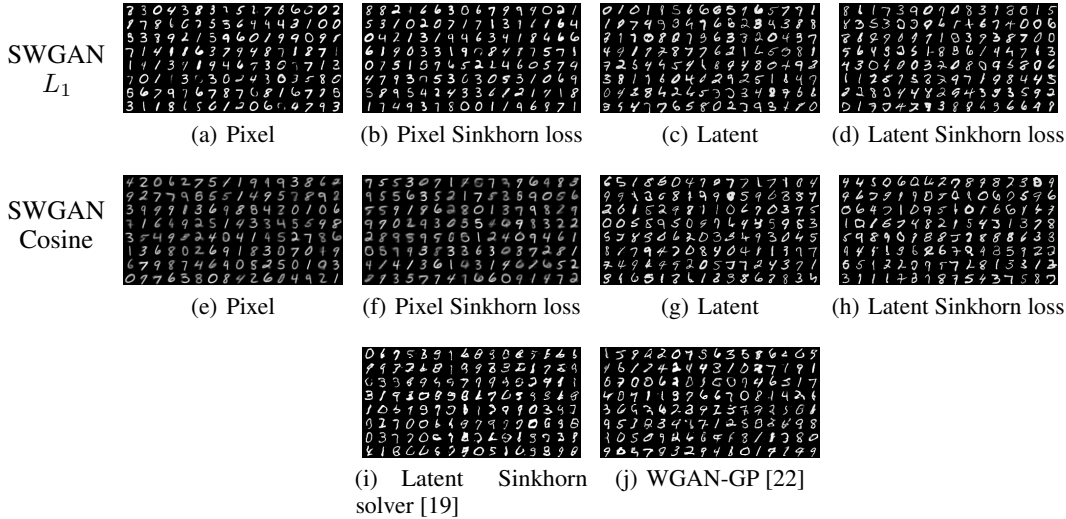

SWGAN
$L_1$

(a) Pixel      (b) Pixel Sinkhorn loss      (c) Latent      (d) Latent Sinkhorn loss

SWGAN
Cosine

(e) Pixel      (f) Pixel Sinkhorn loss      (g) Latent      (h) Latent Sinkhorn loss

(i) Latent Sinkhorn solver [19]      (j) WGAN-GP [22]

Figure 1: Generated MNIST samples using different SWGAN and benchmark methods

by a Convolution Neural Network. As proposed by [41, 18], this latent representation could be adversarially trained to improve the quality of the final results; see Appendix L.1 for more details.

Comparing SWGAN methods with and without latent representation, we find that the ones with latent representation perform better. This might be due to the existence of many bad local minima in the high dimensional pixel space cost function which the algorithm cannot avoid. In contrast, the ones with lower dimensional latent representations seem to have easier time avoiding such bad local minima when the latent representation is also being updated. Based on Lemma 4.3, we conjecture that in these cases, ground truth is the only solution that is stationary regardless of the latent representation. Thus, updating the latent representation once in a while prevents the generative parameters from converging to local minima, i.e. over-fitting to a specific representation. Note that this conjecture is not a direct result of Lemma 4.3.

It is also interesting to note the difference between Fig. 1 (a) and (f), where (f) outperforms (a) which produces many faint images. We believe that the change of objective from regularized OT to Sinkhorn loss helps the method find a better stationary solution, which is closer to the underlying ground truth as predicted by Lemma 4.3. This difference is more pronounced in the experiments on CIFAR-10.

We have also included samples generated by other methods [18, 22] in the last row of Fig. 1. Compared to these methods, specially [18] which uses Sinkhorn algorithm to solve regularized OT, SWGAN methods are capable of generating higher quality images. We also noted that SWGAN methods qualitatively converge faster than other methods. We will formalize this comparison in the experiments on CIFAR-10 using the inception score [40].

## 5.2 Generating tiny color images

To further investigate the performance of SWGAN, we apply it to model 32x32 color images from CIFAR-10 [26]. We compare the SWGAN approach with WGAN-GP [22], OT-GAN [41] and Sinkhorn solver [18]. All the methods are trained using the same architecture and batch-size of 150; see Appendix M for the details and a list of hyper-parameters. We use inception score [40] to compare the quality of generated samples. Learning CIFAR-10 images is a more challenging problem than MNIST; and as we predicted in Section 4.1 SWGAN methods with regularized OT objective cannot generate high quality samples, even with carefully tuned hyper-parameters; see Fig. 4 in Appendix. Due to high computational cost, we only evaluate latent Sinkhorn loss SWGAN with $L_1$ and Cosine cost on CIFAR-10. As can be seen in Fig. 2 (c), given the same architecture and computational power, the SWGAN methods have faster convergence compared to other algorithms. In Fig. 2 all the methods have been running for roughly the same amount of time. Note that OT-GAN [41] is slower as it uses two batches for each label, i.e. fake and real, and requires more computations. We also depict samples of the generated images by SWGAN methods in Fig. 2 (a) and (b). We noted that

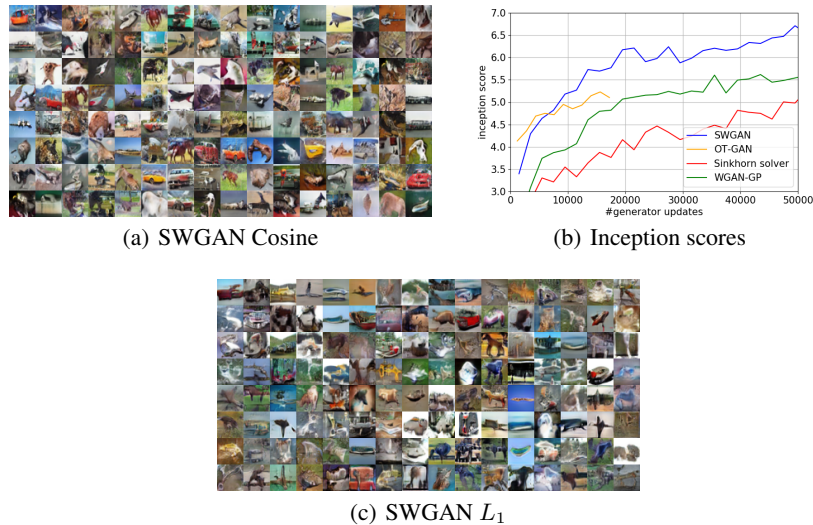

(a) SWGAN Cosine

(b) Inception scores

(c) SWGAN $L_1$

Figure 2: Generated CIFAR-10 samples and inception scores.

SWGAN with $L_1$ cost converges faster than the cosine distance in terms of inception scores, but the samples from the cosine distance model are more visually appealing than the ones from $L_1$.

### Acknowledgments

MS and JDL acknowledge support from ARO W911NF-11-1-0303. The authors would like to thank the anonymous reviewers whose comments/suggestions helped improve the quality/clarity of the paper.

## Footnotes

[1] Throughout the paper we have the hidden technical assumption that $G_\theta$ is a one-to-one mapping on $\mathcal{X}$. This is a reasonable assumption since the space of $\mathcal{Y}$ is usually a low dimensional manifold in higher dimensional space which could be approximated by a mapping of low dimensional code words $x \in \mathcal{X}$. Therefore, the mappings are going to be from low dimensions to high dimensions.

[2]A function $f(\theta) : \mathbb{R}^d \to \mathbb{R}$ is said to be $L$-Lipschitz smooth if it is differentiable and its derivative is $L$-Lipschitz, i.e. $\forall \theta_1, \theta_2 : \|\nabla f(\theta_1) - \nabla f(\theta_2)\| \le L\|\theta_1 - \theta_2\|$.

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
