[Supplementary Material]

**Algorithm 2** Oracle based Non-Convex SGD for GANs with Sinkhorn loss

---

INPUT: $\mathbf{q}, \mathbf{p}, \lambda, S, \theta_0, \{\alpha_t > 0\}_{t=0}^{T-1}$
**for** $t = 0, \cdots, T-1$ **do**
    Call the oracle to find $\epsilon$-approximate maximizer $(\phi_t^1, \psi_t^1)$ for the dual formulation of $d_{c,\lambda}(G_\theta(\mathbf{q}), \mathbf{p})$
    Call the oracle to find $\epsilon$-approximate maximizer $(\phi_t^2, \psi_t^2)$ for the dual formulation of $d_{c,\lambda}(G_\theta(\mathbf{q}), G_\theta(\mathbf{q}))$
    Sample I.I.D. points $x_t^1, \cdots, x_t^S \sim \mathbf{q}, y_t^1, \cdots, y_t^S \sim \mathbf{p}$
    Sample I.I.D. points $\bar{x}_t^1, \cdots, \bar{x}_t^S \sim \mathbf{q}, \hat{x}_t^1, \cdots, \hat{x}_t^S \sim \mathbf{q}$
    Compute
$$g_t = \frac{2}{S^2} \sum_{i,j} \frac{\pi_t^1(G_\theta(x_t^i), y_t^j)}{\mathbf{q}(x_t^i)\mathbf{p}(y_t^j)} \nabla_\theta c(G_\theta(x_t^i), y_t^j) - \frac{1}{S^2} \sum_{i,j} \frac{\pi_t^2(G_\theta(\bar{x}_t^i), G_\theta(\hat{x}_t^i))}{\mathbf{q}(\bar{x}_t^i)\mathbf{q}(\hat{x}_t^i)} \nabla_\theta c(G_\theta(\bar{x}_t^i), G_\theta(\hat{x}_t^i)),$$

    where $\pi_t^i$ is computed using $(\phi_t^i, \psi_t^i)$ based on (6) for $i = 1, 2$.
    Update $\theta_{t+1} \leftarrow \theta_t - \alpha_t g_t$
**end for**

---

# A    Pseudo-distance property of $d_{c,\lambda}$

[13] proves that a constrained version of (4) is a pseudo-distance on the space of probabilities, i.e it is symmetric, non-negative and satisfies the triangular inequality, when $c$ is a proper distance. We prove a similar result for the regularized distance (4).

**Theorem A.1.** *If $c$ is a proper distance, then $d_{c,\lambda}$, defined in* (4) *is a pseudo-distance, i.e. it is non-negative, symmetric, and satisfies the triangular inequality.*

*Proof.* The proof of this theorem is very similar to [13], which proves the result for the constrained version of this objective instead of the regularized one. $d_{c,\lambda}$ is obviously symmetric and non-negative as $c$ is proper norm. In order to prove the triangular inequality, let us take three random variables $X, Y$ and $Z$. Now assume that $\pi_1$ and $\pi_2$ are the transports that achieve the minimum $d_{c,\lambda}$ between $X$ and $Z$ and $Y$ and $Z$, i.e. $\pi_1 = \arg\min_{\pi \in \Pi(X,Z)} H(\pi)$ and $\pi_2 = \arg\min_{\pi \in \Pi(Y,Z)} H(\pi)$. Now construct $\pi(x,y) = \int_z \frac{\pi_1(x,z)\pi_2(y,z)}{p(z)} dz$. It is easy to verify that $\pi(x,y) \in \Pi(X,Y)$. Moreover,

$$\int_x \int_y \pi(x,y)c(x,y)$$
$$= \int_x \int_y \int_z \frac{\pi_1(x,z)\pi_2(y,z)}{p(z)} c(x,y) \, dz \, dx \, dy$$
$$\leq \int_x \int_y \int_z \frac{\pi_1(x,z)\pi_2(y,z)}{p(z)} (c(x,z) + c(y,z)) \, dz \, dy \, dx$$
$$= \int_z \int_x \frac{\pi_1(x,z)c(x,z)}{p(z)} \underbrace{\int_y \pi_2(y,z) \, dy}_{=p(z)} dx \, dz + \int_z \int_y \frac{\pi_2(y,z)c(y,z)}{p(z)} \underbrace{\int_x \pi_1(x,z) \, dx}_{=p(z)} dy \, dz$$
$$= d_{c,\lambda}(X,Z) - \lambda I(\pi_1; X, Z) + d_{c,\lambda}(Y,Z) - \lambda I(\pi_1; Y, Z),$$

where $I(\pi; A, B)$ means the mutual information between $A$ and $B$ when their joint distribution is $\pi$. Now in order to finish the proof, we just need to prove that $I(\pi; X, Y) \leq I(\pi_1; X, Z) + I(\pi_2; Y, Z)$ in order to guarantee $d_{c,\lambda}(X,Y) \leq d_{c,\lambda}(X,Z) + d_{c,\lambda}(Y,Z)$. In fact, we prove a stronger result that $I(\pi; X, Y) \leq \min(I(\pi_1; X, Z), I(\pi_2; Y, Z))$. With some abuse of notation, let us define $p(x,y,z) = \frac{\pi(x,z)\pi(y,z)}{p(z)}$. It is obvious that $\pi(x,y) = \int_z p(x,y,z) \, dz$. Now think of $p$ as a joint distribution for $X, Y$ and $Z$. It is easy to verify that $p(x,y,z)$ could be written as

$$p(x,y,z) = p(x)p(z|x)p(y|z) = p(y)p(z|y)p(x|z). \tag{11}$$

Therefore, with joint distribution $p$, $X$ and $Y$ are independent given $Y$, i.e. $X \to Z \to Y$ and its reverse are both Markov chains. Thus, based on the data processing inequality $I(\pi; X, Y) \leq \min(I(\pi_1; X, Z), I(\pi_2; Y, Z))$. $\qquad\square$

# B  Uniform convergence of $d_{c,\lambda}$ to $d_c$

We state and prove the following uniform convergence between $d_{c,\lambda}$ and $d_c$ which is very similar to the one in concurrent work [7].

**Lemma B.1.** *For any $\lambda \geq 0$,*

$$d_{c,\lambda}(G_\theta(\mathbf{p}), \mathbf{q}) - \lambda I_\theta^* \ \leq \ d_c(G_\theta(\mathbf{p}), \mathbf{q}) \ \leq \ d_{c,\lambda}(G_\theta(\mathbf{p}), \mathbf{q}), \tag{12}$$

*where $I_\theta^*$ is defined as*

$$
\begin{aligned}
I_\theta^* = \min_{\pi \in \Pi(\mathbf{q},\mathbf{p})} \ &I(\pi) \\
\text{s.t. } \ &\mathbb{E}_{x,y \sim \pi}[c(G_\theta(x), y)] \leq d_c(G_\theta(\mathbf{q}), \mathbf{p}).
\end{aligned} \tag{13}
$$

*Proof.* The proof of the rightmost inequality is a simple consequence of the fact that $I(\pi) \geq 0$. The proof of the other inequality uses the fact that $I_\theta^*$ is bounded. Thus, one can plug in the optimal transport plan into the regularized objective and get an upper-bound for the regularized optimal transport. □

In the case of discrete random variables with finite support, we can bound $I_\theta^*$ when $I$ is the KL divergence or norm-2:

**Corollary B.1.1.** *Assume $\mathbf{q}$ and $\mathbf{p}$ represent uniform discrete measures with supports of size $M$ and $N$ respectively. Let us define $K = \min(M, N)$, then*

- *if $I$ is the KL divergence, then $I_\theta^* \leq \log(K)$*

- *if $I$ is the norm-2 regularizer, then $I_\theta^* \leq \frac{K}{2}$*

*Proof.* The proof for the KL divergence is due to the fact that $I(\pi) \leq \min(\text{Entropy}(\mathbf{p}), \text{Entropy}(\mathbf{q}))$ [12]. For the case of norm-2 regularization, the proof is a simple consequence of the fact that if $\pi$ marginalizes to $\mathbf{p}$ and $\mathbf{q}$, then none of its entries could be larger than $\frac{1}{\max(M,N)}$. □

The above bound is a pessimistic bound and, in fact, the two distances might be closer in reality depending on the distributions. On the other hand, in the continuous setting, $I_\theta^*$ could be infinite. Therefore, obtaining a uniform bound on the difference between the two distances is impossible, a point-wise convergence between the two distances as $\lambda \to 0$ can be obtained [42].

# C  Proof of Theorem 3.1

*Proof.* Let us first prove the result for the case where $\mathcal{X}$ and $\mathcal{Y}$ have finite supports. While the proof of the general case requires a more careful application of Danskin's theorem, it will essentially follow the same steps.

In the finite support case, differentiability is a consequence of applying a simple version of Danskin's theorem, e.g. [6, Theorem 0], and the uniqueness of the minimzer. To apply Danskin's theorem, we need the set $\Pi(\mathbf{p}, \mathbf{q})$ to be compact which is obvious in the finite support case. Moreover, in this case the differentiability and strong convexity of $H$ is evident. Thus, the uniqueness of the minimizer follows immediately and the differentiability could be deduced based on Dankin's Theorem, i.e. [6, Theorem 0].

Now that we have the differentiability, we can focus on the rest of the proof. For any $\theta_1$ and $\theta_2$, let us define $\pi_i^* = \arg\min_{\pi \in \Pi(\mathbf{p},\mathbf{q})} H(\pi, \theta_i)$, $i = 1, 2$. Due to the optimality of $\pi^*$, we have $\langle \nabla_\pi H(\theta, \pi^*(\theta)), \pi - \pi^*(\theta) \rangle \geq 0$ for all feasible $\pi$. Due to the strong convexity of $H$ with respect to $\pi$ we have:

$$H(\theta_2, \pi_2^*) \geq H(\theta_2, \pi_1^*) + \langle \nabla_\pi H(\theta_2, \pi_1^*), \pi_2^* - \pi_1^* \rangle + \frac{\lambda}{2}\|\pi_1^* - \pi_2^*\|_1^2 \tag{14}$$

$$H(\theta_2, \pi_1^*) \geq H(\theta_2, \pi_2^*) + \underbrace{\langle \nabla_\pi H(\theta_2, \pi_2^*), \pi_1^* - \pi_2^* \rangle}_{\geq \, 0, \text{ due to optimality of } \pi_2^*} + \frac{\lambda}{2}\|\pi_1^* - \pi_2^*\|_1^2 \tag{15}$$

Moreover, due to optimality of $\pi_1^*$, we have
$$\langle \nabla_\pi H(\theta_1, \pi_1^*), \pi_2^* - \pi_1^* \rangle \geq 0 \tag{16}$$
Adding up all these inequalities, we get
$$\langle \nabla_\pi H(\theta_2, \pi_1^*) - \nabla_\pi H(\theta_1, \pi_1^*), \pi_1^* - \pi_2^* \rangle \geq \lambda \|\pi_1^* - \pi_2^*\|_1^2 \tag{17}$$
Now we use the holder inequality and get
$$\lambda \|\pi_1^* - \pi_2^*\|_1 \leq \|\nabla_\pi H(\theta_2, \pi_1^*) - \nabla_\pi H(\theta_1, \pi_1^*)\|_\infty. \tag{18}$$
Note that $\nabla_\pi(\theta, \pi) = c(G_\theta(x), y) + \lambda(1 + \log(\pi(x, y)))$. Therefore, it is easy to see that
$$\|\nabla_\pi H(\theta_2, \pi_1^*) - \nabla_\pi H(\theta_1, \pi_1^*)\|_\infty$$
$$\leq \sup_{x,y} |c(G_{\theta_1}(x), y) - c(G_{\theta_2}(x), y)|$$
$$\leq L_0 \|\theta_1 - \theta_2\|, \tag{19}$$
where the last inequality is due to the assumption. This proves the fact that $\|\pi_1^* - \pi_2^*\| \leq \frac{L_0}{\lambda} \|\theta_1 - \theta_2\|$.

To prove the Lipschitz smoothness, note that based on Danskins' theorem
$$\nabla h_\lambda(\theta) = \nabla_\theta d_{c,\lambda}(G_\theta(\mathbf{p}), \mathbf{q}) = \mathbb{E}_{x,y \sim \pi^*(\theta)}\left[\nabla_\theta c(G_\theta(x), y)\right]$$

Therefore, for any two parameters $\theta_1$ and $\theta_2$,
$$\|\nabla_\theta d_{c,\lambda}(G_{\theta_1}(\mathbf{p}), \mathbf{q}) - \nabla_\theta d_{c,\lambda}(G_{\theta_2}(\mathbf{p}), \mathbf{q})\| \leq$$
$$\|\mathbb{E}_{x,y \sim \pi_1^*}[\nabla_\theta c(G_{\theta_1}(x), y)] - \mathbb{E}_{x,y \sim \pi_2^*}[\nabla_\theta c(G_{\theta_1}(x), y)]\|$$
$$+ \|\mathbb{E}_{x,y \sim \pi_2^*}[\nabla_\theta c(G_{\theta_1}(x), y)] - \mathbb{E}_{x,y \sim \pi_2^*}[\nabla_\theta c(G_{\theta_2}(x), y)]\|$$
$$\leq L_0 \|\pi_1^* - \pi_2^*\|_1 + L_1 \|\theta_1 - \theta_2\| \leq \left(L_1 + \frac{L_0^2}{\lambda}\right)\|\theta_1 - \theta_2\|, \tag{20}$$
where the inequalities are based on triangle inequality, assumptions and the stability result we just proved respectively.

Now that we have proven the results for distributions with finite support, we are ready to point out the changes in the proof steps for obtaining the same results for continuous distributions. Notably, most of the changes are related to the application of Danskin't theorem and obtaining differentiability.

In order to prove differentiability for continuous case we use a more suitable version of Danskin's Theorem, i.e. Theorem D1 in [6]. In order to apply this result we require $\Pi(\mathbf{p}, \mathbf{q})$ to be compact and $H$ to be lower semi-continuous w.r.t $\pi$. In fact, $\Pi(\mathbf{p}, \mathbf{q})$ is compact under narrow topology (see Definition 1.2 in [10]) when $\mathcal{X}$ and $\mathcal{Y}$ are compact and $\mathbf{p}$ and $\mathbf{q}$ have bounded Entropy; see Proposition 2.3 in [11] or Lemma 4.4 in [44].

Also note that the function $H$ could be decomposed into two parts, i.e.
$$H(\pi, \theta) = \underbrace{\mathbb{E}_{(x,y) \sim \pi} c(G_\theta(x), y)}_{C(\pi, \theta)} + \underbrace{KL(\pi \,\|\, \mathbf{p} \times \mathbf{q})}_{I(\pi)} \tag{21}$$
It is obvious that $C(\pi, \theta)$ is linear in $\pi$ and therefore continuous. Moreover, $I(\pi)$ is also lower-semi-continuous w.r.t. $\pi$ in narrow topology; see Theorem 1 in [38]. It is worth noting that the rest of the technical conditions for Danskin, i.e. Theorem D1 in [6], are also satisfied. This is because $C(\pi, \theta)$, which is the only term in $H$ containing $\theta$, is differentiable and has bounded and continuous directional derivatives based on our assumptions on $c$ and $G_\theta$.

Also note that $\pi(x, y) = \mathbf{q}(x)\mathbf{p}(y)$ is always admissible in the minimization and achieves a finite cost due to the assumptions. Therefore, $\inf_{\pi \in \Pi(\mathbf{p}, \mathbf{q})} H(\pi, \theta)$ is always attained due to the compactness of $\Pi(\mathbf{p}, \mathbf{q})$ and moreover at any optimal point $I$ is bounded; see Proposition 2.3 in [10]. Note that as $I$ is bounded for any optimal solution of the minimization, they all have to be absolutely continuous w.r.t. base measure $\mathbf{q}(x) \times \mathbf{p}(y)$ [38]. By applying Danskin's Theorem we know that
$$Dh_\lambda(\theta; d) = \max_{\pi \in \pi^*(\theta)} \mathbb{E}_{(x,y) \sim \pi} \nabla_\theta c(G_\theta(x), y)^T d, \tag{22}$$
where $Dh_\lambda(\theta; d)$ is the directional derivative of $h_\lambda$ at point $\theta$ and direction $d$ and $\pi^*(\theta)$ is the set of minimizers for $\min_{\pi \in \Pi(\mathbf{p}, \mathbf{q})} H(\pi, \theta)$. Now we invoke the following result to show that $\pi^*(\theta)$ has unique solution with respect to total variation norm or $\ell_1$ distance of probability distributions. The proof of this Lemma is presented in Section N for the sake of completeness.

**Lemma C.1.** $I(\pi) = KL(\pi\|\mu)$ *is strongly convex with modulus 1 with respect to total variation norm, i.e.* $d(\nu, \xi) = \int |\nu - \xi| = \|\nu - \xi\|$, *on the set of $\pi$'s where $\pi$ is absolutely continuous w.r.t. $\mu$.*

Based on Lemma C.1, for any $\pi, \hat{\pi} \in \pi^*(\theta)$, $\|\pi - \hat{\pi}\|_1 = 0$. Thus, for any $\pi \in \pi^*(\theta)$ we get the same directional derivative expression $\mathbb{E}_{(x,y)\sim\pi}\nabla_\theta c(G_\theta(x), y)^T d$ for any direction $d$. As a result, with some abuse of notation we can write $Dh_\lambda(\theta; d) = \mathbb{E}_{(x,y)\sim\pi^*(\theta)}\nabla_\theta c(G_\theta(x), y)^T d$. Thus, $h_\lambda$ is differentiable and $\nabla h_\lambda(\theta) = \mathbb{E}_{(x,y)\sim\pi^*(\theta)}\nabla_\theta c(G_\theta(x), y)$.

The key to continue the rest of the proof steps for continuous distributions is to use strong-convexity, i.e. Lemma C.1 along-side directional derivatives. In order to carry out the proof correctly, we need to prove a result similar to (17), i.e.

$$\langle \nabla_\pi C(\theta_2, \pi_1^*) - \nabla_\pi C(\theta_1, \pi_1^*), \pi_1^* - \pi_2^* \rangle \geq \lambda \|\pi_1^* - \pi_2^*\|_1^2 \tag{23}$$

To do so, we need to use intermediate steps similar to (14), (15) and (16). Note that as $\pi_1^*$ and $\pi_2^*$ are both absolutely continuous with respect to $\mathbf{q} \times \mathbf{p}$ (due to their optimality), $H(\pi, \theta)$ is strongly convex in $\pi$ on the line segment between $\pi_1^*$ and $\pi_2^*$. To obtain results similar to (14), (15) and (16) we also use the existance of directional derivative of $H$ w.r.t. $\pi$, i.e.

$$D_\pi H(\theta, \pi_i^*; \pi_j^* - \pi_i^*) = \langle \nabla_\pi C(\theta, \pi_i^*), \pi_j^* - \pi_i^* \rangle + DI(\pi_i^*; \pi_j^* - \pi_i^*), \tag{24}$$

where $DI(\pi_i^*; \pi_j^* - \pi_i^*)$ is the directional derivative of $I$ at $\pi_i^*$ in direction $\pi_j^* - \pi_i^*$ which is bounded. Thus, we can write counterparts of (14), (15) and (16) in continuous case as follows

$$H(\theta_2, \pi_2^*) \geq H(\theta_2, \pi_1^*) + \langle \nabla_\pi C(\theta_2, \pi_1^*), \pi_2^* - \pi_1^* \rangle + DI(\pi_1^*; \pi_2^* - \pi_1^*) + \frac{\lambda}{2}\|\pi_1^* - \pi_2^*\|_1^2 \tag{25}$$

$$H(\theta_2, \pi_1^*) \geq H(\theta_2, \pi_2^*) + \underbrace{\langle \nabla_\pi C(\theta_2, \pi_2^*), \pi_1^* - \pi_2^* \rangle + DI(\pi_2^*; \pi_1^* - \pi_2^*)}_{\geq\, 0,\text{ due to optimality of } \pi_2^*} + \frac{\lambda}{2}\|\pi_1^* - \pi_2^*\|_1^2 \tag{26}$$

$$\langle \nabla_\pi C(\theta_1, \pi_1^*), \pi_2^* - \pi_1^* \rangle + DI(\pi_1^*; \pi_2^* - \pi_1^*) \geq 0 \quad \text{(due to optimality of } \pi_1^*) \tag{27}$$

Note that (23) could be easily obtained by combining the above inequalities. And the rest of the proof is exactly the same as the finite support case.

$\square$

# D   Dual formulation for regularized optimal transport

## D.1   Proof of Lemma 2.1

In this section we prove the results in Lemma 2.1 for the KL regularized optimal transport. The results for 2-norm regularized optimal transport are very similar and could be derived using the same logic.

*Proof.* Let us rewrite the primal problem for optimal transport with function $c(x, y)$ and KL regularizer.

$$\min_\pi \ \int_\mathcal{X} \int_\mathcal{Y} \pi(x, y) c(x, y) dx dy +$$

$$\lambda \int_\mathcal{X} \int_\mathcal{Y} \pi(x, y) \log\left(\frac{\pi(x, y)}{\mathbf{p}(y)\mathbf{q}(x)}\right) dx dy,$$

$$\text{s.t.} \int_\mathcal{X} \pi(x, y) dx = \mathbf{p}(y) \,, \ \int_\mathcal{Y} \pi(x, y) dy = \mathbf{q}(x) \ \& \ \pi(x, y) \geq 0, \tag{28}$$

Now let us introduce the functions $\psi(y)$ and $-\phi(x)$ as the Lagrange multipliers for the two equality constraints respectively. Therefore, we can form the Lagrangian as follows

$$
\min_{\pi(x,y)\geq 0} \int_{\mathcal{X}}\int_{\mathcal{Y}} \Bigg( c(x,y)\pi(x,y)
$$
$$
+ \lambda\pi(x,y)\log\left(\frac{\pi(x,y)}{\mathbf{p}(y)\mathbf{q}(x)}\right)
$$
$$
+ \psi(y)\big(\pi(x,y)-\mathbf{p}(y)\big)
$$
$$
- \phi(x)\big(\pi(x,y-\mathbf{q}(x))\big)\Bigg)dxdy \tag{29}
$$

So the optimality condition for each $\pi(x,y)$ is given as

$$
c(x,y) + \lambda\log\left(\pi^*(x,y)/\mathbf{q}(x)\mathbf{p}(y)\right) + \lambda + \psi(y) - \phi(x) \geq 0, \tag{30}
$$

where the strict inequality holds only if $\pi^*(x,y) = 0$. But if $c(x,y) < \infty$ and $\psi$ and $\phi$ are bounded at each point, $\pi^*(x,y) > 0$. So the equality holds under the assumption that $c(x,y) < \infty$ for all $x \in \mathcal{X}, y \in \mathcal{Y}$. Based on these assumptions, the optimal $\pi^*(x,y)$ is

$$
\pi^*(x,y) = \mathbf{q}(x)\mathbf{p}(y)\exp\left[\frac{\phi(x)-\psi(y)-c(x,y)}{\lambda}-1\right] \tag{31}
$$

Now if we define, $V(x,y) = \phi(x) - \psi(y) - c(x,y)$, we can rewrite the Lagrangian after plugging in the value of $\pi^*$ as

$$
\max_{\psi,\phi}\ \mathbb{E}_{x\sim\mathbf{q}}[\phi(x)] - \mathbb{E}_{y\sim\mathbf{p}}[\psi(y)] - \frac{\lambda}{e}\mathbb{E}_{x,y\sim\mathbf{q}\times\mathbf{p}}\left[e^{\frac{V(x,y)}{\lambda}}\right] \tag{32}
$$

$\square$

### D.2 Solving the dual: parametric vs. non-parametric

Let us focus on the case where the distributions $\mathbf{q}$ and $\mathbf{p}$ are discrete with supports of size $M$ and $N$ respectively. In this case, with some abuse of notation we can define $q_i = \mathbf{q}(x_i)$, $p_i = \mathbf{p}(y_i)$, $c_{ij} = c(x_i, y_j)$, $\phi_i = \phi(x_i)$, $\psi_j = \psi(y_j)$, $\Phi = [\phi_1, \cdots, \phi_M]^T$ and $\Psi = [\psi_1, \cdots, \psi_N]^T$. Thus, the dual formulation of regularized optimal transport could be written as the following unconstrained concave maximization problem

$$
F^* = \max_{\Phi,\Psi}\ F(\Phi,\Psi), \tag{33}
$$
$$
\text{where}\quad F(\Phi,\Psi) = \sum_i q_i\phi_i - \sum_j p_j\psi_j - \sum_{i,j} q_i p_j f_\lambda(\phi_i - \psi_j - c_{ij}).
$$

Here $f_\lambda$ is a convex function which depends on the regularizer and is defined in Lemma 2.1. One way to solve such a problem is to employ convex optimization algorithms. For example, one can employ a first order method, such as (stochastic) gradient descent to solve this problem [18]. Thus, one can find an $\epsilon$-accurate solution $(\hat{\Phi}, \hat{\Psi})$, such that

$$
F^* - F(\hat{\Phi}, \hat{\Psi}) \leq \epsilon, \tag{34}
$$

by applying gradient descent with $O(\frac{1}{\epsilon})$ iterations [36]. While this iterative procedure on this non-parametric problem is guaranteed to converge within a known number of iterations, we cannot benefit from the relationship between $x_i$'s and $y_j$'s. This becomes specially important if $\mathbf{p}$ and $\mathbf{q}$ are empirical distributions of unknown continuous distributions. In those cases, exploiting such relationships by using parametric methods, such as neural networks, can help us obtain discriminators that generalize better. In addition, exploiting such spatial relationships could also result in lower computation and memory usage. Note that for example, each gradient computation for the convex formulation requires $O(M \times N)$ computation which would be cost prohibitive. On the other hand, the parametric approaches such as neural networks could result in much lower complexities at the cost of having little theoretical guarantee. Although, we have empirically observed that parametric methods such as neural networks are capable of solving these problems efficiently.

It is also worth mentioning that Theorem 3.1 shows certain stability of the optimal regularized transport plan under small perturbations of the parameter $\theta$. Similar result can be established for optimal dual variables $(\Phi, \Psi)$; see Appendix I. This result suggests that when performing a small update on the generator parameters, solving the new dual optimal transport problem should not be very difficult if we use a warm-start obtained from the previous dual parameters.

### D.3 Verifying the quality of the dual solutions

One advantage of having an unconstrained dual is that if $F^* - F(\Phi, \Psi) \leq \epsilon$, then we have $\|\nabla F(\Phi, \Psi)\|^2 \leq 2L_F\epsilon$, where $L_F$ is the Lipschitz constant for the gradient of $F$. In other words, if $(\Phi, \Psi)$ is approximately optimal, then $\|\nabla F(\Phi, \Psi)\|$ has to be small. This condition is also applicable in the parameterized formulation, i.e., one can look at the norm of the gradient, with respect to functional values, to decide when to stop the iterative procedure for solving the discriminator problem. However, in practice obtaining this gradient is computationally expensive. In what follows, we suggest an alternative simple verifiable necessary condition to be used as the termination criterion for the discriminator iterative solver. Notice that for any $x_i$, the gradient of $F$ with respect to the corresponding $\phi_i$ is given by $\nabla_{\phi_i} F(\Phi, \Psi) = \mathbf{q}(x_i) - \sum_j \mathbf{q}(x_i)\mathbf{p}(y_j)m_{ij}$, where $m_{ij} = m(x_i, y_j) = \frac{\pi(x_i, y_j)}{\mathbf{q}(x_i)\mathbf{p}(y_j)}$ and $\pi$ is the pseudo transport plan generated by $(\Phi, \Psi)$ using (6). Thus, in order for the gradient to be small, we need

$$\mathbb{E}_{y_j \sim \mathbf{p}} m_{ij} \approx 1. \tag{35}$$

By repeating this argument for $y_j$, we similarly obtain $\mathbb{E}_{x_i \sim \mathbf{q}} m_{ij} \approx 1$. Hence, one probabilisticnecessary condition for these two approximate inequalities to be true is that for IID samples $(x_1, y_1), \cdots, (x_S, y_S)$, we must have

$$\frac{1}{S} \sum_{k=1}^{S} m(x_k, y_k) \approx 1. \tag{36}$$

This condition is true with high probability if sample size $S$ is large enough and $m_{ij}$'s are bounded. We can use this simple measure as a dynamic termination criterion to decide when to stop the iterative discriminator optimizer.

## E Smoothness of 2-norm regularized objective

The following Theorem states the smoothness result for the 2-norm regularized optimal transport and its proof is very similar to the the proof of Theorem 3.1 and thus omitted.

**Theorem E.1.** *If there exists non-negative constants $L_1$, $\hat{\ell}_0$ and $p_{\max}$ and $q_{\max}$, such that:*

- *For any feasible $\theta_1, \theta_2$,*

$$\sup_{x,y} \|\nabla_\theta c(G_{\theta_1}(x), y) - \nabla_\theta c(G_{\theta_2}(x), y)\| \leq L_1 \|\theta_1 - \theta_2\|$$

- *For any feasible $\theta$,*

$$\sqrt{\int_{\mathcal{X}} \int_{\mathcal{Y}} \|\nabla_\theta c(G_\theta(x), y)\|^2} \leq \hat{\ell}_0$$

- $\sup_x \mathbf{q}(x) \leq q_{\max}$ *and* $\sup_y \mathbf{p}(y) \leq p_{\max}$

*then the 2-norm regularized objective $h_\lambda(\theta) = d_{c,\lambda}(G_\theta(\mathbf{p}), \mathbf{q})$ is L-Lipschitz smooth, where $L = L_1 + \frac{\hat{\ell}_0^2 p_{\max} q_{\max}}{\lambda}$. Moreover, for any two parameters $\theta_1$ and $\theta_2$, $\|\pi^*(\theta_1) - \pi^*(\theta_2)\|_2 \leq \frac{\hat{\ell}_0 p_{\max} q_{\max}}{\lambda} \|\theta_1 - \theta_2\|$. In the rest of the paper it is easier to use*

$$\ell_0 = \hat{\ell}_0 \sqrt{q_{\max} p_{\max}}$$

*instead of $\hat{\ell}_0$ to state the results.*

We can further specialize this theorem to the case of discrete uniform distributions. In such case, $p_{\max}q_{\max} = \frac{1}{MN}$.

**Remark E.1.1.** *Assume uniform discrete distributions* $\mathbf{q}$ *and* $\mathbf{p}$ *with supports of size* $M$ *and* $N$ *respectively, then under the assumptions of Theorem E.1, the 2-norm regularized function* $h_\lambda(\theta)$ *is* $L = L_1 + \frac{\ell_0^2}{\lambda}$ *Lipschitz smooth, where*

$$\ell_0 = \frac{\hat{\ell}_0}{\sqrt{MN}}.$$

*Moreover, for any two parameters* $\theta_1$ *and* $\theta_2$, $\|\pi^*(\theta_1) - \pi^*(\theta_2)\|_2 \leq \frac{\ell_0}{\lambda\sqrt{MN}}\|\theta_1 - \theta_2\|$. *As a result* $\|\pi^*(\theta_1) - \pi^*(\theta_2)\|_1 \leq \frac{\ell_0}{\lambda}\|\theta_1 - \theta_2\|$. *Note that in this case* $\hat{\ell}_0$ *increases at the rate* $\sqrt{MN}$ *with the size while* $\ell_0$ *does not increase when the sizes* $M$ *and* $N$ *increase. That is the reason* $\ell_0$ *is better than* $\hat{\ell}_0$ *to represent the results.*

**Remark E.1.2.** *Note that typically* $\ell_0 \ll L_0$, *which is defined in Theorem 3.1. This means that the norm-2 regularized surrogate is smoother than the KL regularized one, when we use the same* $\lambda$.

# F  Example: $W_2^2$-GANs with linear generator

Let us take a deeper look at the problem for the simple case where $c(z, y) = \frac{1}{2}\|z - y\|^2$, and $X$ is an $n$-dimensional zero mean random variable where $\|X\| \leq r_x$ and $Y$ is a $d$-dimensional zero mean random variable where $\|Y\| \leq r_y$ [17]. Moreover, let us assume that the generator is linear, i.e. we have matrix $\Theta \in \mathbb{R}^{d \times n}$ where $G_\Theta(x) = \Theta x$ and we try to find the best linear generator that maps the distribution of $X$, i.e. $\mathbf{q}$ to the distribution of $Y$, i.e. $\mathbf{p}$. In other words, we try to solve

$$\min_\Theta d_c(\Theta\mathbf{q}, \mathbf{p}), \tag{37}$$

where

$$d_c(\Theta\mathbf{q}, \mathbf{p}) = \min_{\pi \in \Pi(\mathbf{p}, \mathbf{q})} \frac{1}{2} \int_{\mathcal{X}, \mathcal{Y}} \pi(x, y)\|\Theta x - y\|^2 dx dy.$$

$W_2^2$ GAN formulation could be further simplified as

$$\min_\Theta \frac{1}{2}\text{Tr}(\Theta\Sigma_X\Theta^T) + \frac{1}{2}\text{Tr}(\Sigma_Y) + \min_{\pi \in \Pi(\mathbf{p}, \mathbf{q})} \int_{\mathcal{X}, \mathcal{Y}} -x^T\Theta^T y \, \pi(x, y) dx dy, \tag{38}$$

where $\Sigma_X$ and $\Sigma_Y$ are covariances for $\mathbf{q}$ and $\mathbf{p}$ respectively. As we mentioned earlier, the problem in (38) is not convex (because the last part is the min of linear functions) nor smooth (due to the fact that optimal solution of the min might not be unique). Therefore, we can approximate it with a smooth function when we add a strongly convex regularizer to the end of the inside minimization. The following two corollaries are simple applications of our smoothness result in the case of linear generator.

**Corollary F.0.1.** *If* $c(z, y) = \frac{1}{2}\|z - y\|^2$, *then* $h_\lambda(\Theta) = d_{c,\lambda}(\Theta\mathbf{q}, \mathbf{p})$ *is differentiable. Moreover, if* $\forall x \in \mathcal{X}, \|x\|_2 \leq r_x$ *and* $\forall y \in \mathcal{Y}, \|y\|_2 \leq r_y$ *and for any* $\Theta_1$ *and* $\Theta_2$, *if* $\pi_i^* = \arg\min_{\pi \in \Pi} H(\Theta_i, \pi)$, $i = 1, 2$, *then*

$$\|\pi_1^* - \pi_2^*\|_1 \leq \frac{r_x r_y}{\lambda}\|\Theta_1 - \Theta_2\|_F. \tag{39}$$

*Moreover,* $h_\lambda(\theta)$ *is Lipschitz smooth with respect to Frobeneous norm with constant* $L = \sigma_{\max}^2(\Sigma_X) + \frac{r_x^2 r_y^2 \sqrt{nd}}{\lambda}$.

In addition, the following corollary could be easily obtained from the above due to the special structure of the function in the linear case.

**Corollary F.0.2.** *If* $\mathbf{q}$ *is a uniform distribution on the sphere of radius* $r_x$, *then* $L = \frac{r_x^2}{n} + \frac{r_x^2 r_y^2 \sqrt{nd}}{\lambda}$. *Moreover, if* $\lambda \geq nr_y^2\sqrt{nd}$, *then the function* $d_{c,\lambda}$ *would be convex and the minimizer of it would be at* $\Theta = 0$.

Basically, the above corollary says that far away from the optimal solutions, the function has a convex behavior, while the non-convexity occurs when we get close to optimal solutions. It also states that increasing $\lambda$ would cause a bias in the obtained solution towards zero and eventually will make the solution obsolete.

# G  Proof of Theorem 4.1

*Proof.* We prove this theorem for the case where $\mathcal{X}$ and $\mathcal{Y}$ have finite support as it requires less technical assumptions. A concrete general proof for the continuous case requires some technicality which would be beyond the scope of this work. We first prove the theorem for the 2-norm regularizer. Let us define $m(x, y) = \frac{\pi(x,y)}{\mathbf{q}(x)\mathbf{p}(y)}$ and also assume $\mathbf{q}(x) \leq q_{\max}, \ \forall x$ and $\mathbf{p}(y) \leq p_{\max}, \ \forall y$. Note that when $\mathcal{X}$ and $\mathcal{Y}$ have finite support it is obvious that $p_{\max}, q_{\max} \leq 1$. With some abuse of notation, we can see that the dual problem in this case could equivalently be written as

$$\max_{\Phi, \Psi, \pi} \ \mathbb{E}_{x \sim \mathbf{q}} \phi(G_\theta(x)) - \mathbb{E}_{y \sim \mathbf{p}} \psi(y) - \frac{\lambda}{2} \mathbb{E}_{(x,y) \sim \mathbf{q} \times \mathbf{p}} m(x, y)^2$$

$$\text{s.t.} \ \ m(x, y) = \frac{\pi(x, y)}{\mathbf{q}(x)\mathbf{p}(y)} \geq 0, \ m(x, y) \geq \frac{\phi(G_\theta(x)) - \psi(y) - c(G_\theta(x), y)}{\lambda}. \tag{40}$$

This is because by optimizing over $\pi$ when fixing $\Phi$ and $\Psi$, we get the same objective as before. Now if we have a solution $(\Phi, \Psi)$ that is $\epsilon$-accurate, we can form the triple $P = (\Phi, \Psi, \pi)$, where $\pi$ is chosen to be optimal for $(\Phi, \Psi)$ (see (6) and Lemma 2.1 for the exact expression). Obviously, this triple is $\epsilon$-accurate for the alternative optimization problem (40). We can do the same for the optimal solution $(\Phi^*, \Psi^*)$ to get the triple $P^* = (\Phi^*, \Psi^*, \pi^*)$, which would be optimal for (40). Now both of these triples are feasible, and the feasible set is convex, therefore the line segment between them should be feasible. As a result of the optimality of $V^*$, the directional derivative of the objective in (40) at $V^*$ along this feasible direction has to be non-positive. But this means that the inner product of the gradient of the objective with the vector $\Delta P = P - P^*$ is non-positive. In addition, note that the objective is strongly concave in $\pi$ with modulus $\frac{\lambda}{p_{\max} q_{\max}}$, as a result we have the following inequality

$$\frac{\lambda}{2 p_{\max} q_{\max}} \|\pi - \pi^*\|_2^2 \leq \epsilon. \tag{41}$$

Thus, $\|\pi - \pi^*\|_2 \leq \sqrt{\frac{2 p_{\max} q_{\max} \epsilon}{\lambda}}$. Moreover, note that

$$\nabla_\theta h_\lambda = \mathbb{E}_{x,y \sim \pi^*} \left[ \nabla_\theta c(G_\theta(x), y) \right] \tag{42}$$

Now we use the assumptions of Remark E.1.1 on $c$ and Holder inequality to get

$$\left\| \mathbb{E}_{x,y \sim \mathbf{q} \times \mathbf{p}} \left[ \underbrace{\frac{\pi(x, y)}{\mathbf{q}(x)\mathbf{p}(y)}}_{m(x,y)} \nabla_\theta c(G_\theta(x), y) \right] - \nabla_\theta h_\lambda \right\|$$

$$\leq \int_{\mathcal{X}, \mathcal{Y}} |\pi(x, y) - \pi^*(x, y)| \|\nabla_\theta c(G_\theta(x), y)\|$$

$$\leq \hat{\ell}_0 \sqrt{\frac{2 p_{\max} q_{\max} \epsilon}{\lambda}} = \ell_0 \sqrt{\frac{2\epsilon}{\lambda}}, \tag{43}$$

where $\ell_0$ and $\hat{\ell}_0$ are defined in Theorem E.1.

In order to prove the theorem for KL regularizer, let us define $a(x, y) = \sqrt{\pi(x, y)}$. Now let us re-write the dual problem for KL regularized optimal transport using this new set of variables

$$\max_{\Phi, \Psi, a} \ \mathbb{E}_{x \sim \mathbf{q}} \phi(G_\theta(x)) - \mathbb{E}_{y \sim \mathbf{p}} \psi(y) - \lambda \int_{\mathcal{X}, \mathcal{Y}} a(x, y)^2$$

$$\text{s.t.} \ \ a(x, y) \geq \frac{1}{\sqrt{e}} \exp \left( \frac{\phi(G_\theta(x)) - \psi(y) - c(G_\theta(x), y)}{2\lambda} \right). \tag{44}$$

Using the same argument as before, we can form $(\Phi, \Psi, a)$ which is $\epsilon$-optimal, $(\Phi^*, \Psi^*, a^*)$ which is optimal. Similar to the 2-norm regularized case and due to strong convexity with respect to $a$ we have

$$\|a^* - a\|_2^2 \leq \frac{\epsilon}{\lambda} \tag{45}$$

Now, let us define $\pi_{\max} = \max_{x,y}(\max(\pi(x,y), \pi^*(x,y)))$, where $\pi(x,y) = a^2(x,y)$ and $\pi^*(x,y) = (a^*(x,y))^2$. If we use the concavity of the square root function, we can see that $|a(x,y) - a^*(x,y)| \geq \frac{1}{2\sqrt{\pi_{\max}}}|\pi(x,y) - \pi^*(x,y)|$. Therefore,

$$\|\pi - \pi^*\|_2 \leq 2\sqrt{\pi_{\max}\frac{\epsilon}{\lambda}}.$$

As a result with the same argument and assumption as before we get

$$\left\|\mathbb{E}_{x,y\sim\mathbf{q}\times\mathbf{p}}\left[\frac{\pi(x,y)}{\mathbf{q}(x)\mathbf{p}(y)}\nabla_\theta c(G_\theta(x),y)\right] - \nabla_\theta h_\lambda\right\| \leq 2\hat{\ell}_0\sqrt{\pi_{\max}\frac{\epsilon}{\lambda}} = 2\ell_0\sqrt{\frac{\pi_{\max}}{p_{\max}q_{\max}}\frac{\epsilon}{\lambda}},$$

Note that $\max_{(}x,y)(\pi^*(x,y)) \leq \min(p_{\max}, q_{\max})$ as $\pi^*$ has $\mathbf{p}$ and $\mathbf{q}$ as its marginals. Moreover, due to (45) $a(x,y) \leq a^*(x,y) + \sqrt{\frac{\epsilon}{\lambda}}$. Thus, $\max_{x,y}(a(x,y)) \leq \max_{x,y} a^{(}x,y) + \sqrt{\epsilon}\lambda$. Squaring both sides and using the inequality $(\alpha + \beta)^2 \leq 2(\alpha^2 + \beta^2)$ we get

$$\pi_{\max} \leq 2\left(\min(p_{\max}, q_{\max}) + \frac{\epsilon}{\lambda}\right)$$

$\square$

**Remark G.0.1.** *In practice, we have noticed that when the scale of the problem increases the KL regularizer does not produce very good results. This could be partially explained by the above result and how $\frac{\pi_{\max}}{p_{\max}q_{\max}}$ would scale with the problem size. Note that when $\mathbf{q}$ and $\mathbf{p}$ are uniform distributions with support $M$ and $N$ respectively, then $\pi_{\max} \leq 2\min(M,N) + 2MN\frac{\epsilon}{\lambda}$ which could be large depending on $\frac{\epsilon}{\lambda}$.*

## H  Proof of Theorem 4.2

*Proof.* Let us define some notations to make the proof more readable. Let us define $\nabla_t = \nabla h_\lambda(\theta_t)$ and $\mathbb{E}(g_t|\pi_t) = G_t$. Note that based on Theorem 4.1, $\|G_t - \nabla_t\| \leq \delta$.

Due to the smoothness of $d_{c,\lambda}$, which we proved in Theorem 3.1, and the update rule $\theta_{t+1} = \theta_t - \alpha_t g_t$

$$h_\lambda(\theta_{t+1}) \leq h_\lambda(\theta_t) + \langle\nabla_t, \theta_{t+1} - \theta_t\rangle + \frac{L}{2}\|\theta_{t+1} - \theta_t\|^2$$

$$= h_\lambda(\theta_t) - \alpha_t\langle\nabla_t, g_t\rangle + \frac{L\alpha_t^2}{2}\|g_t\|^2$$

$$= h_\lambda(\theta_t) - \frac{\alpha_t}{2}\left(\|\nabla_t\|^2 + \|g_t\|^2 - \|\nabla_t - g_t\|^2\right) + \frac{L\alpha_t^2}{2}\|g_t\|^2$$

Now we replace $g_t$ with $g_t - G_t + G_t$ in all the expressions and use $\|a+b\|^2 = \|a\|^2 + \|b\|^2 + 2\langle a, b\rangle$. After re-arranging the terms we get

$$h_\lambda(\theta_{t+1}) \leq h_\lambda(\theta_t) - \frac{\alpha_t}{2}\|\nabla_t\|^2 + \left(\frac{L\alpha^2 - \alpha}{2}\right)\|G_t\|^2$$

$$+ \frac{\alpha}{2}\|\nabla_t - G_t\|^2 + \frac{L\alpha^2}{2}\|G_t - g_t\|^2 - \alpha\langle\nabla_t, g_t - G_t\rangle + L\alpha^2\langle G_t, g_t - G_t\rangle \tag{46}$$

Now we are ready to sum up the inequalities across the iterations to get

$$\frac{\alpha}{2}\sum_{t=1}^T\|\nabla_t\|^2 \leq \overbrace{h_\lambda(\theta_0) - h_\lambda(\theta_{T+1})}^{\leq\Delta}$$

$$+ \sum_{t=1}^T\left[\left(\frac{L\alpha^2 - \alpha}{2}\right)\|G_t\|^2 + \frac{\alpha}{2}\|\nabla_t - G_t\|^2 + \frac{L\alpha^2}{2}\|G_t - g_t\|^2\right.$$

$$\left. - \alpha\langle\nabla_t, g_t - G_t\rangle + L\alpha^2\langle G_t, g_t - G_t\rangle\right] \tag{47}$$

We want to take the expectation of both sides of the inequality over the randomness in the choices of $g_t$. Let us define the history up iteration $t$ as $\xi_t = [\theta_0, \cdots, \theta_t, \pi_0, \cdots, \pi_t]$. Note that $\mathbb{E}[g_t - G_t | \xi_t] = 0$, while $\nabla_t$ and $G_t$ are fixed given $\xi_t$. Therefore, by conditioning on the history while taking expectations the last two terms in the sum would be zero. In addition, as a consequence of Theorem 4.1, $\|\nabla_t - G_t\| \leq \delta$. Moreover, we have assumed $\mathbb{E}[\|G_t - g_t\|^2 | \xi_t] \leq \sigma^2$. Therefore, the final inequality after taking the expectations would be

$$\frac{\alpha}{2} \sum_{t=1}^{T} \mathbb{E}[\|\nabla_t\|^2] \leq \Delta + T\left(\frac{\alpha\delta^2}{2} + \frac{L\alpha^2\sigma^2}{2}\right) + \frac{L\alpha^2 - \alpha}{2} \sum_{t=1}^{T} \mathbb{E}[\|G_t\|^2] \tag{48}$$

Now we consider two different scenarios:

- If number of iterations $T$ is large enough, i.e. $T \geq \frac{2\Delta L}{\sigma^2}$, then by choosing $\alpha_t = \alpha = \sqrt{\frac{2\Delta}{TL\sigma^2}}$, we have that $\frac{L\alpha^2 - \alpha}{2} \leq 0$. Thus, the last sum on the right hand side of (48) non-positive. Therefore, we have

$$\frac{1}{T} \sum_{t=1}^{T} \mathbb{E}[\|\nabla_t\|^2] \leq \sqrt{\frac{8\Delta L \sigma^2}{T}} + \delta^2.$$

- If the number of iterations is too small, i.e. $T < \frac{2\Delta L}{\sigma^2}$, then we choose $\alpha_t = \alpha = \frac{1}{L}$. In a such a case, $\frac{L\alpha^2 - \alpha}{2} = 0$. Thus, we have

$$\frac{1}{T} \sum_{t=1}^{T} \mathbb{E}[\|\nabla_t\|^2] \leq \frac{2L\Delta}{T} + \delta^2 + \sigma^2$$

In this case, as $T$ cannot grow to infinity, the right hand side is bounded below by $2\sigma^2 + \delta^2$.

Note that the first regime is the interesting as it results in asymptotic convergence rate of expected norm of gradient as $T \to \infty$. Therefore, it is the one that is mentioned in the body of Theorem 4.2. □

# I Stability of dual variables

As part of Theorem 3.1 we proved the stability of the transport plan $\pi$ under small perturbations in the generator parameters $\theta$. But there are infinitely many dual variables $(\Phi, \Psi)$ corresponding to each transport plan. Therefore, it is not completely clear if the dual variables would also be stable, i.e. while we perform small changes in the generator parameter $\theta$, how far do we need to change $(\Phi, \Psi)$ to get to an optimal one for the new parameter $\theta$. The following theorem characterizes such a stability in the case of KL regularizer.

**Theorem I.1.** *Let us assume that we use the KL regularizer with weight $\lambda$ and $\mathbf{q}$ and $\mathbf{p}$ have finite supports. For any two generator parameters $\theta$, $\theta'$, define $\pi = \pi^*(\theta)$ and $\pi' = \pi^*(\theta')$, and $(\Phi, \Psi)$ as a set of optimal dual parameters corresponding to $\pi$. Let us further assume that $\pi, \pi' \geq \pi_{\min}$, i.e. the transport plan probabilities are bounded away from zero. Then there exists $(\Phi', \Psi')$, which corresponds to $\pi'$ and*

$$\sqrt{\|\Phi' - \Phi\|_2^2 + \|\Psi' - \Psi\|_2^2} \leq O\left(\frac{\|\theta - \theta'\|}{\pi_{\min}}\right) \tag{49}$$

**Remark I.1.1.** *The above theorem proves that when we purturb $\theta$, finding a new dual solution, starting from an old solution for the old $\theta$ should be generally easy while $\pi_{\min}$ is bunded away from zero. But it also suggests that such a problem will get more difficult when $\pi_{\min}$ goes down, i.e. when the two distributions get closer to each other or $\lambda$ is very small.*

*Proof.* Let us define $c_{ij} = c(G_\theta(x_i), y_j)$ and $c'_{ij} = c(G_\theta(x_i), y_j)$. Also we assume that the support of $\mathbf{q}$ and $\mathbf{p}$ are of sizes $M$ and $N$ respectively. From Theorem 3.1 we know that $\|\pi - \pi'\|_1 \leq$

$\frac{L_0}{\lambda}\|\theta - \theta'\|$. Due to the mapping (6) and strong convexity of exponential function on bounded domain we have

$$\pi_{\min} \sum_{i,j} |\phi_i - \psi_j - c_{ij} - \lambda \log(e\pi'_{ij})| \leq L_0 \|\theta - \theta'\| \tag{50}$$

We also know that $|c_{ij} - c'_{ij}| \leq L_0 \|\theta - \theta'\|$. As a result

$$\sum_{i,j} |\phi_i - \psi_j - c'_{ij} - \lambda \log(e\pi'_{ij})|$$

$$\leq \sum_{i,j} |\phi_i - \psi_j - c_{ij}| + |c_{ij} - c'_{ij}|$$

$$\leq \left( \frac{L_0}{\pi_{\min}} + L_0 MN \right) \|\theta - \theta'\|$$

$$\leq L_0 (1 + \pi_{\min} MN) \frac{\|\theta - \theta'\|}{\pi_{\min}} \tag{51}$$

The above result shows that $(\Phi, \Psi)$ approximately satisfy the system linear equations, i.e.

$$\phi_i - \psi_j \approx c'_{ij} + \lambda \log(e\pi'_{ij}), \quad \forall (i,j).$$

It is obvious that any accurate solution of that system of linear equations would be an optimal dual solution corresponding to $\pi'$. To find such a solution with the minimum distance to $(\Phi, \Psi)$ we just need to project $(\Phi, \Psi)$ to the subspace of all solutions of the above system of linear equations. If we call the projected point by $(\Phi', \Psi')$, the it is obviously a set of dual variables corresponding to $\pi'$. Moreover, using basic linear algebra we can see that

$$\sqrt{\|\Phi' - \Phi\|_2^2 + \|\Psi' - \Psi\|_2^2} \leq \frac{1}{\|A\|_2} \|d\|_2, \tag{52}$$

where $\|A\|_2$ is the operator norm of the linear operator corresponding to the above linear system of equation and $\|d\|$ is the norm of the violation. From (51) we know that

$$\|d\|_2 \leq L_0 (1 + \pi_{\min} MN) \frac{\|\theta - \theta'\|}{\pi_{\min}}.$$

Moreover, it can be easily verified that $\|A\|_2 = \sqrt{M + N}$ for the above linear system of equation. Plugging everything back we get

$$\sqrt{\|\Phi' - \Phi\|_2^2 + \|\Psi' - \Psi\|_2^2} \leq \frac{L_0 (1 + \pi_{\min} MN)}{\sqrt{M + N}} \frac{\|\theta - \theta'\|}{\pi_{\min}}$$

Note that if $\mathbf{q}$ and $\mathbf{p}$ are uniform, then it is meaningful to assume that $\pi_{\min} = \frac{c}{MN}$ for some $c \leq 1$[3]. With this conversion, the bound becomes

$$\sqrt{\|\Phi' - \Phi\|_2^2 + \|\Psi' - \Psi\|_2^2} \leq L_0 \frac{(c+1)MN}{c\sqrt{M + N}} \|\theta - \theta'\|. \tag{53}$$

$\square$

**Remark I.1.2.** *Note that in equation (53) even if we control for the size of the vectors $(\Phi, \Psi)$, the stability result is getting worse linearly by when we increase $M$ and $N$. This seems to be the downside of using the dual formulation as the dual solutions will be farther apart when the number of points increases. This is in contrary with the primal stability result in which the stability is irrespctive of the number of data points.*

## J  Proof of Lemma 4.3

*Proof.* Note that in $L_\lambda(G_\theta(\mathbf{p}), \mathbf{q})$ the last term does not have any gradient with respect to $\theta$. Thus,

$$\nabla_\theta L_\lambda(G_\theta(\mathbf{p}), \mathbf{q}) = 2\nabla_\theta \bar{d}_{c,\lambda}(G_\theta(\mathbf{p}), \mathbf{q}) - \nabla_\theta \bar{d}_{c,\lambda}(G_\theta(\mathbf{p}), G_\theta(\mathbf{p})). \tag{54}$$

When $\theta = \theta_0$, we have $G_\theta(\mathbf{p}) = G_{\theta_0}(\mathbf{p}) = \mathbf{q}$. Thus, the unique optimal transport plan for $\bar{d}_{c,\lambda}(G_\theta(\mathbf{p}), G_{\theta_0}(\mathbf{p}))$ and $\bar{d}_{c,\lambda}(G_\theta(\mathbf{p}), G_\theta(\mathbf{p}))$ are the same regardless of the choice of $\lambda$. Let us denote this transport plan by $\pi^*$. Furthermore, we have:

$$\nabla_\theta \bar{d}_{c,\lambda}(G_\theta(\mathbf{p}), G_{\theta_0}(\mathbf{p})) = \int\int J_\theta(x)\nabla_1 c(G_\theta(x), G_{\theta_0}(\hat{x}))d\pi^*,$$

$$\nabla_\theta \bar{d}_{c,\lambda}(G_\theta(\mathbf{p}), G_\theta(\mathbf{p})) = \int\int J_\theta(x)\nabla_1 c(G_\theta(x), G_\theta(\hat{x}))d\pi^* +$$
$$\int\int J_\theta(\hat{x})\nabla_2 c(G_\theta(x), G_\theta(\hat{x}))d\pi^*,$$

where $\nabla_i c$ is the gradient of $c$ with respect to its $i$-th input, $i = 1, 2$. Moreover, $J_\theta(x)$ is the Jacobian of $G_\theta(x)$ with respect to $\theta$. Note that $x$ and $\hat{x}$ come from the same distribution $\mathbf{p}$.

Now, note that if $c$ is symmetric, as $G_\theta(\mathbf{p}) = G_{\theta_0}(\mathbf{p})$ the transport plan $\pi^*(x, \hat{x})$ is symmetric too. As $c$ is symmetric $\nabla_1 c(y, \hat{y}) = \nabla_2 c(\hat{y}, y)$. As a result if $\theta = \theta_0$

$$\int\int J_\theta(x)\nabla_1 c(G_\theta(x), G_\theta(\hat{x}))d\pi^* = \int\int J_\theta(\hat{x})\nabla_2 c(G_\theta(x), G_\theta(\hat{x}))d\pi^* \qquad (55)$$

This means that $\nabla_\theta \bar{d}_{c,\lambda}(G_\theta(\mathbf{p}), G_\theta(\mathbf{p}))\big|_{\theta=\theta_0} = 2\nabla_\theta \bar{d}_{c,\lambda}(G_\theta(\mathbf{p}), G_{\theta_0}(\mathbf{p}))\big|_{\theta=\theta_0}$. This directly implies that $\nabla_\theta L_\lambda(G_\theta(\mathbf{p}), \mathbf{q})\big|_{\theta=\theta_0} = 0$. $\qquad\square$

## K  Recovering mixture of 2D Gaussians on a grid

We first apply SWGAN on a simple synthetic data set of mixture of 25 Gaussians proposed by [28] to show the convergence of all the modes. We first use the setup proposed in [28], which uses $d = 4$ dimensional Gaussian code; for details on the architecture of the networks and hyper-parameters see Appendix L.2. Figure 3 shows the result of running our smoothed WGAN (SWGAN) algorithm for 10,000 generator iterations. The ground truth points are depicted in red while the generated points are in blue. As can be seen in the figure, our method has perfectly recovered all the modes. As a benchmark we also included the results of running WGAN-GP algorithm [22] with a set of manually tuned hyper-parameters. Note that WGAN-GP is one of the most stable algorithms in the literature. Similar to our method, WGAN-GP softly imposes 1-Lipschitzness by regularizing the objective.

(a) SWGAN $d = 4$    (b) WGAN-GP $d = 4$    (c) SWGAN $d = 2$    (d) WGAN-GP $d = 2$

Figure 3: Learning mixture of 2-D Gaussian using random codes of dimension $d = 4$ and $d = 2$: (a) output of SWGAN with code dimension $d = 4$ after 10,000 iterations ($\approx$ 13 mins run time on a machine with one K-80 gpu) (b) output of WGAN-GP with code dimension $d = 4$ after 30,000 iterations ($\approx$ 14 mins run time on the same machine) (c) output of SWGAN with code dimension $d = 2$ after 10,000 iterations (d) output of WGAN-GP with code dimension $d = 2$ after 30,000 iterations

In order to make the problem more challenging and show the robustness of our method, we reduce the size of the random codes to $d = 2$. We use our method with the exact same hyper parameters and setup. With $d = 2$ the WGAN-GP solution quality deteriorates substantially. Figure 3 shows the results for our SWGAN as well as the best result we obtained with WGAN-GP with $d = 2$ in the same amount of time. Note that our algorithm robustly identifies all the modes even in this challenging setup for which WGAN-GP does not perform very well.

Table 1: Hyper-parameters for training mixture of 2D Gaussians.

| | |
|---|---:|
| $\lambda$ | 0.01 |
| GEN. LEARNING RATE | 0.003 |
| BATCH SIZE | $128 \times 128$ |
| DISC. LEARNING RATE | 0.001 |
| REGULARIZER | 2-NORM |
| ADAM PARAMETERS | $\beta_1 = 0.5, \beta_2 = 0.9$ |
| #DISC ITERS/GEN ITER | 20 |
| #GEN ITERS | $10,000$ |
| $c(x, y)$ | $\|x - y\|_1$ |

## L  Training details

### L.1  Learning data-dependent cost function

We can construct a more meaningful latent representation for images and apply $c$ in a latent space $\eta_\gamma(\cdot)$ parameterized by a set of weights $\gamma$. Note that our convergence analysis applies for a fixed representation. In practice, one can fix the representation for a few iterations and update it once in a while in adversarial fashion so that it does not allow the method to converge to bad local minima. This idea has been around and many different ways for updating this representation adversarially has been proposed [5, 41, 19]. In our experiments we found that the best results would be obtained when we update the representation by applying a gradient step on parameters $\gamma$ based on the following adversariall objective:

$$\max_\gamma \bar{L}_\lambda(\gamma) = 2d_{c,\lambda}(\eta_\gamma(G_\theta(\mathbf{p})), \eta_\gamma(\mathbf{q})) + d_{c,\lambda}(\eta_\gamma(G_\theta(\mathbf{p})), \eta_\gamma(G_\theta(\mathbf{p}))) + d_{c,\lambda}(\eta_\gamma(\mathbf{q}), \eta_\gamma(\mathbf{q})).$$

(56)

This heuristic adversarial training of the data-dependent cost functions promotes the mapping to be more informative in differentiating between samples in the original/fake distribution as well as between fake and real distributions. We believe this would lead to learning more meaningful representations. The standard training procedure for GAN is to alternate the discriminator and the generator update. When learning the additional data-dependent $c(\eta_\gamma(x), \eta_\gamma(y))$, we update the cost function once every few generator updates. In our experiment, we used a ratio of 5 generator updates to one cost function update.

### L.2  Training details for learning mixture of 2D Gaussians

Similar to [28] we use a generator with two fully connected hidden layers, each of which with 128 neurons with $\tanh$ activation. In [28] authors propose to use a discriminator with two fully connected hidden layers of 128 neurons and ReLU activation. For WGAN-GP we use one full discriminator with hidden layers of size 128 neurons.

In each generator iteration, we performed at most 20 iterations of discriminator, while dynamically checking the optimality condition of (36) to stop the update of discriminator.

Table 1 summarizes the chosen hyper-parameters for our method.

### L.3  Training details for MNIST digits

Table 2 summarizes the list of hyper parameters used in our training for MNIST data set.

We use a model architectures similar to DCGAN [39]. A 128 dimensional standard multivariate Gaussian is passed to a fully connected layer of 4096 hidden units. This is followed by three deconvolutional layers to generate the final 28x28 image. The discriminator has three convolutional layers using 64, 128, 256 filters with stride of 2. The feature map from the last convolutional layer is then flattened out to produce the discriminator output with a linear layer. leaky ReLU non-linearity are used in both generator and discriminators. Batch normalization and Adam are used in both generator and discriminator. The structures for our generator and discriminator networks are summerized below:

Table 2: Hyper-parameters for training digits on MNIST.

| $\lambda$ | 0.5 |
|---|---|
| GEN. LEARNING RATE | 2E-4 |
| BATCH SIZE | 100 |
| DISC. LEARNING RATE | 2E-4 |
| ADAM PARAMETERS | $\beta_1 = 0., \beta_2 = 0.95$ |
| #DISC ITERS/GEN ITER | 5 |
| #GEN ITER/ADVERSARIAL DISTANCE | 2 |
| #GEN ITERS | 50000 |
| $c(x,y)$ | $\|x - y\|_1$ OR $1 - \frac{x^T y}{\|x\|_2 \|y\|_2}$ |

Table 3: Hyper-parameters for training CIFAR-10 images.

| $\lambda$ | 0.5 |
|---|---|
| GEN. LEARNING RATE | 2E-4 |
| BATCH SIZE | 150 |
| DISC. LEARNING RATE | 2E-4 |
| ADAM PARAMETERS | $\beta_1 = 0., \beta_2 = 0.95$ |
| #DISC ITERS/GEN ITER | 3 |
| #GEN ITER/ADVERSARIAL DISTANCE | 2 |
| #GEN ITERS | 50000 |
| $c(x,y)$ | $\|x - y\|_1$ OR $1 - \frac{x^T y}{\|x\|_2 \|y\|_2}$ |

- Generator: [ FC(128, 4096)-BN-ReLU-DECONV(256, 128,stride=2)-BN-ReLU-DECONV(128, 64,stride=2)-BN-ReLU-DECONV(128, 1,stride=2)-Tanh ]

- Discriminator: [ [ CONV(1, 64,stride=2)-BN-LReLU-CONV(64, 128,stride=2)-BN-LReLU-CONV(128, 256,stride=2)-BN-LN-LReLU]-FC(4096, 5)-BN-LReLU-FC(5, 1) ]

- Adversarially learnt CNN feature denoted using []

# M   Training details for CIFAR

We use the following structures for our generator and discriminator networks as [41]:

- Generator: [ FC(128, 16384)-BN-GLU-UpSample2x-CONV(1024, 512,stride=1)-BN-GLU-UpSample2x-CONV(512, 256,stride=1)-BN-GLU-UpSample2x-CONV(256, 128,stride=1)-BN-GLU-CONV(128, 3,stride=1)-Tanh ]

- Discriminator: [ [CONV(3, 256,stride=1)-BN-CReLU-CONV(256, 512,stride=2)-BN-CReLU-CONV(64, 1024,stride=2)-BN-CReLU-CONV(128, 2048,stride=2)-BN-LN-CReLU]-FC(32768, 5)-BN-LReLU-FC(5,1) ]

- Adversarially learnt CNN feature denoted using []

# N   Strong convexity of KL for continuous distributions

In this section we re-state the strong convexity result of Lemma C.1. For the sake of completeness, we provide a proof for this Lemma. This simple and elegant proof was proposed in [*].

**Lemma N.1.** *$I(\pi) = KL(\pi\|\mu)$ is strongly convex with modulus 1 with respect to total variation norm, i.e. $d(\nu, \xi) = \int |\nu - \xi| = \|\nu - \xi\|_1$, on the set of $\pi$'s where $\pi$ is absolutely continuous w.r.t. $\mu$.*

*Proof.* We use the following definition of strong convexity

**Definition N.1.1.** *Function $I$ is strongly convex with modulus $\sigma$ if for any $\pi_0$ and $\pi_1$, we have*

$$\Delta_t = (1 - t)I(\pi_0) + tI(\pi_1) - I(\pi_t) \geq \sigma \frac{(1 - t)t}{2} \|\pi_0 - \pi_1\|^2, \tag{57}$$

*where $\pi_t = t\pi_1 + (1 - t)\pi_0$.*

Figure 4: Results of learning CIFAR-10 images using regularized OT with manually chosen hyper-parameters and carefully annealing regularization weight: (a) These blurry samples are the best samples we could get with SWGAN with regularized OT (b) The algorithm starts becoming unstable as we anneal the regularization weight to get sharper images (c) The algorithm finally diverges and produces garbage as the discriminator loses it accuracy. (d) directly learning in the pixel space with Sinkhorn loss and fixed regularization weight.

Let us assume that $\pi_1$ and $\pi_0$ are absolutely continuous with respect to $\mu$. Define $f_t = \frac{\pi_t}{\mu}$. By Taylor's theorem with the integral form of the remainder[4] for $h(x) := x \ln x$ we have

$$h(f_j) = h(f_t) + h'(f_t)(f_j - f_t) + (f_j - f_t)^2 \int_0^1 h''((1-s)f_t + sf_j)(1-s)\,ds, \quad \text{for } j = 0, 1.$$

Thus,

$$\begin{aligned}
\delta &:= (1-t)h(f_0) + th(f_1) - h(f_t) \\
&= (1-t)t\,(f_1 - f_0)^2 \int_0^1 \left( \frac{t}{(1-s)f_t + sf_0} + \frac{1-t}{(1-s)f_t + sf_1} \right)(1-s)\,ds \\
&= (1-t)t\,(f_1 - f_0)^2 \int_0^1 \left( \frac{t}{f_{u_0(t,s)}} + \frac{1-t}{f_{u_1(t,s)}} \right)(1-s)\,ds,
\end{aligned}$$

where

$$u_j(t,s) := (1-s)t + js.$$

So,

$$\Delta_t = \int \delta\mu = (1-t)t \int_0^1 (1-s)\,ds\,[tJ(u_0(t,s)) + (1-t)J(u_1(t,s))], \qquad (58)$$

where

$$J(u) := \int \frac{(f_1 - f_0)^2}{f_u}\,dQ.$$

Next, take any $u \in (0,1)$. Then $\pi_1$ is absolutely continuous w.r. to $\pi_u$. Introducing $g_u := \dfrac{\pi_1}{\pi_u} = \dfrac{f_1}{f_u}$, we have

$$
\begin{aligned}
J(u) &= \frac{1}{(1-u)^2} \int \frac{(f_1 - f_u)^2}{f_u} \mu \\
&= \frac{1}{(1-u)^2} \int (g_u - 1)^2 \pi_u \\
&\geq \frac{1}{(1-u)^2} \left( \int |g_u - 1| \pi_u \right)^2 = \frac{1}{(1-u)^2} \|\pi_1 - \pi_u\|_1^2 = \|\pi_1 - \pi_0\|_1^2.
\end{aligned} \tag{59}
$$

Note also that for any $t \in (0,1)$ and $s \in (0,1)$ we have $u_0(t,s) \in (0,1)$ and $u_1(t,s) \in (0,1)$ and hence, by (59), $J(u_j(t,s)) \geq \|\pi_1 - \pi_0\|_1^2$ for $j = 0, 1$. This completes the proof. □

## Footnotes

[3]That is because $\pi_{ij} = \frac{1}{MN}$ is equivalent to randomly assigning the two points

[4] see https://www.math.umd.edu/ jmr/141/remainder.pdf

## References

[*] Iosif Pinelis *"https://mathoverflow.net/users/36721/iosif-pinelis"* Is KL divergence $D(P\|Q)$ strongly convex over $P$ in infinite dimension. *MathOverflow,* *"https://mathoverflow.net/q/307251".*