[Reviews · NeurIPS 2018]

Reviewer 1



SUMMARY The authors investigate the task of training a Generative Adversarial Networks model based on optimal transport (OT) loss. They focus on regularized OT losses, and show that approximate gradients of these losses can be obtained by approximately solving regularized OT problem (Thm 4.1). As a consequence, a non-convex stochastic gradient method for minimizing this loss has a provable convergence rate to stationarity (Thm 4.2). The analysis also applies to Sinkhorn losses. The authors then explore numerically the behavior of a practical algorithm where the dual variable are parametrized by neural networks (the theory does not immediately apply because estimating the loss gradient becomes non-convex). QUALITY I find that the point of view adopted by the authors is relevant. Training GAN is known to suffer from instabilities and here the authors provide theoretical tools to understand how to obtain gradients of the loss with a bounded error. While this relation between regularization and stability is not new in optimization/statistics, it seems to be invoked with proofs in this specific application for the first time. Yet, the authors provide numerical results on simple datasets which are interpretable. CLARITY The article is well written, well organized, and easy to read. ORIGINALITY The theoretical results are new in this setting but rely on well known principles. The numerical algorithm is, I believe, a minor variation of previously proposed Wasserstein GAN algorithms. SIGNIFICANCE The paper does not provide a strikingly new point of view, but make reliable progress in the direction of robust training for GANs. SPECIFIC REMARKS - l.130 "for practical considerations the dual functions belong to the set of parametric functions": this is likely to considerably change the (geometrical and statistical) properties of the distance and is instrumental. I would be in favor that the community as a whole emphasize more on the differences between these "parametrized Wasserstein" and the classical Wasserstein distances (I just state this as an opinion). - l.167 "although pi may not be a feasible transport plan": it is feasible, as a solution to Eq.(4). - l.174 again, I think the switch to parametric dual function cannot be reduced to a mere computational trick; - theorem 3.1 it seems that there is redundancy in the second and third conditions. The definition of L-Lipschitz smooth could maybe be given? - proof of Th.3.1 (in Appendix C): why is the 1-norm of plans appearing (Jensen's inequality should be explicitly invoked I guess) (l.449 typo after nabla). - l.200 typo "the its" - l.272 I'd say that neural network falls also in the class of parametric approach. - l.273 I didn't understand why gradients are unbiased, is there any result supporting this claim ? This is a very important claim. - l.292 this is a very (!) strong conjecture... in any case, Lemma 4.3 is not supporting this conjecture since it is just a sufficient stationarity condition. - l.299 again, it is not predicted by Lemma 4.3 which does not characterize stationary points in general. [UPDATE AFTER REBUTTAL I have read the reviews and authors' response. As reviewer #2, I thought that the setting was that of arbitrary probability measures (including "continuous" marginals with densities), as implied by the notations of the paper. I am not entirely convinced by the comment in author's response stating that "discrete distributions are the case in practice where distributions are represented by their samples" because the smoothness should hold for the "expected" risk, although the algorithm only uses samples. Looking at the proof, it is true that it lacks some details required to correctly deal with the continuous case (details to apply Danskin’s theorem, choices of topologies). Some technical difficulties might appear although I do not expect the results to be different. The authors in the rebuttal claim that they can adapt the proof to fix these issues. Therefore, I still lean towards acceptance although I am slightly less convinced than before, because of this confusion between continuous and discrete setting.]

Reviewer 2



Update: After reading the author feedback, I think the paper provides a good insight on the properties of regularized OT in the context of learning and the proofs seem correct. The paper would be stronger if the results also included continuous distributions. ----------------------------------------------------------------- ------------------------------------------------------------------------------------------------- The paper provides theoretical guarantees for the convergence of GANs based on regularized Wasserstein distance. The approach itself was already proposed in (Seguy et. al 2017), so the main contribution of the paper is really the theory. The three key results are the following: the authors show that the regularized Wasserstein distance is smooth with respect to the parameters of the generator. They provide a bound on the error when approximately computing the gradient of the distance, where the error comes from solving the dual problem up to some precision. They finally show that the expected gradient averaged over the iterations remains within a ball which radius depends on the accuracy of the approximation of the distance. The paper is globally well written although some notations are not properly introduced. Overall, the contribution seems rather marginal as the results are somehow expected, especially theorems 4.1 and 4.2. They rely on the fact that the dual problem can be solved up to some accuracy epsilon. However, there is no guarantee this precision is ever achieved when using a fixed size neural network to approximate the dual solution. On the other hand, theorem 3.1 could be more useful, but there are some points that need to be clarified: - In the proof of theorem 3.1, it is not straightforward to me how Danskin's theorem applies to this case. The most common version of the theorem requires that the set under which the optimization occurs to be compact which seems not to be the case for the set of all possible couplings \pi. Since theorem 3.1 seems to be one of the main contributions of this work, it would be good to know precisely how the assumptions are used to prove the differentiability and which version of Danskin's theorem is used with proper references. -Also, the set of joint distributions seems to be endowed with some norm, is it the total variation norm on signed measures? I'm guessing it is rather the L_{1} norm under dp(x)dp(y)? which would make sense since the optimal coupling is known to have a density with respect to dp(x)dq(y) by lemma 2.1, however, the notation is rather confusing at first sight. - In theorem 4.1: The statement is a priory for arbitrary probability distribution p and q, however the proof in the appendix assumes that p and q are discrete and relies on extra assumptions stated in Remark E.1.1. It seems however that a more general proof could be obtained using only assumptions of theorem 3.1: inequalities similar to 31 and 35 could be obtained for any p and q under an appropriate norm.

Reviewer 3



This paper considers training Generative Adversarial Networks (GAN) using the technique of Regularized Optimal Transport, shows the convergence of this method, and also considers a variant with more robust performance. In particular, it considers regularized Wasserstein distance for training the generative model. It first shows that this objective is smooth w.r.t. parameters in the generative model, then shows that an approximate solution for the discriminator allows to approximately compute the (stochastic) gradient of the generator, and finally shows the convergence of SGD/GD to an approximate stationary point where the solution quality depends on the average error of the discriminator steps. The work then points out that in practice a small regularizer is needed but will lead to computational challenges, and proposes to use a more robust objective Sinkhorn loss (the convergence analysis also applies here). The method leads to promising empirical results on standard datasets MNIST and CIFAR-10. Pros: 1. The work proposes a simple and effective method for training GAN, with solid theoretical analysis. The work adopts techniques from several lines of existing work and the proposed framework is clean and mathematically well formulated. It also provides an analysis showing global convergence to approximate stationary points and points out factors affecting the approximation guarantee, building on top of the smoothness from the regularization. 2. The experimental results seem promising. In particular, the inception score of the proposed method on CIFAR-10 is well above those of several existing methods. Cons: 1. The convergence is to (approximate) stationary points. It is unclear whether the stationary points are good solutions from the generative model perspective and under what conditions such a guarantee can be achieved. But I understand that this is beyond the scope of this work.